

# Effective field theory of fluctuating wall in open systems: from a kink in Josephson junction to general domain wall

**Keisuke Fujii[1][⋆] and Masaru Hongo[2,3][†]**

**1** Institut für Theoretische Physik, Universität Heidelberg, D-69120 Heidelberg, Germany
**2** Department of Physics, Niigata University, Niigata 950-2181, Japan
**3** RIKEN iTHEMS, RIKEN, Wako 351-0198, Japan

⋆ fujii@thphys.uni-heidelberg.de, † hongo@phys.sc.niigata-u.ac.jp

## Abstract

We investigate macroscopic behaviors of fluctuating domain walls in nonequilibrium open systems with the help of the effective field theory based on symmetry. Since the domain wall in open systems breaks the translational symmetry, there appears a gapless excitation identified as the Nambu-Goldstone (NG) mode, which shows the non-propagating diffusive behavior in contrast to those in closed systems. After demonstrating the presence of the diffusive NG mode in the $(2 + 1)$-dimensional dissipative Josephson junction, we provide a symmetry-based general analysis for open systems breaking the one-dimensional translational symmetry. A general effective Lagrangian is constructed based on the Schwinger-Keldysh formalism, which supports the presence of the gapless diffusion mode in the fluctuation spectrum in the thin wall regime. Besides, we also identify a term peculiar to the open system, which possibly leads to the instability in the thick-wall regime or the nonlinear Kardar-Parisi-Zhang coupling in the thin-wall regime although it is absent in the Josephson junction.



# 1 Introduction

A domain wall, codimension one object, is ubiquitous in nature from the condensed-matter physics to the high-energy physics [1–5]. The sine-Gordon kink in the Josephson junction [6], the magnetic domain wall in various magnets [7], the interface of two different phases [8] separated by, e.g., the first-order phase transition (like liquid and gas), and extended membrane-like objects in the string theory [9] all give the domain-wall realization in diverse physical systems. There are several different reasons why the domain wall is a stable object appearing in diverse systems: for instance, some of the domain walls are topological solitons [4,5] showing a particle-like behavior, and others have the topological charge supporting its stability.

A remarkable property of the domain-wall solution is that it breaks the one-dimensional spatial translational symmetry. As a result, a fluctuation of the domain-wall position propagates as a gapless mode in closed systems. The presence of the propagating gapless mode is universal independent of the underlying microscopic model, and this gapless mode is identified as the Nambu-Goldstone (NG) mode [10–12] associated with the translational symmetry breaking. One way to describe the universal macroscopic dynamics of the domain wall is to use the low-energy effective field theory (EFT) based on the symmetries [13–20]. Recent progresses in the nonrelativistic generalization of the NG theorem enables us to establish a sophisticated EFT approach to the domain-wall dynamics based on the spacetime symmetry breaking [21] and also to provide a unified view on the coupled dynamics of the domain wall and other NG modes [22,23].

Turning our attention to nonequilibrium systems, we find qualitatively different domain-wall dynamics from the aforementioned gapless propagating mode. In nonequilibrium open systems, we often encounter the gapless *diffusive* fluctuation instead of the gapless *propagating* one. A familiar example of the domain-wall dynamics is a linear surface growth between two different phases, which provides an example of the universality class in nonequilibrium systems modeled by the Edwards-Wilkinson equation [24]. Besides, there is another universality class driven by nonlinear fluctuations, which leads to the so-called the Kardar-Parisi-Zhang

(KPZ) universality class [25]. Recent experimental and theoretical developments have demonstrated the presence of the KPZ universality class in various low-dimensional systems [26–32]. Nevertheless, despite these developments, the universal domain-wall dynamics in nonequilibrium open systems has not been clarified so far in a unified way with those in closed systems based on the symmetry-based EFT.

The main purpose of this paper is twofold: The first purpose is to elucidate the low-energy dynamics of the magnetic flux in the (2+1)-dimensional dissipative Josephson junction, which gives a canonical condensed-matter example of the domain wall. The Josephson junction is a layer insulator sandwiched by two superconducting electrodes, for which the dynamics of the phase difference between the two electrodes is modeled by the sine-Gordon equation (see, e.g., Refs. [6,33]). Moreover, the dissipative effects due to the environment (such as electrons or phonons) are inevitable in experimental finite-temperature realizations, and the effective description is given by the dissipative generalization of the sine-Gordon equation [33–37]. Thus, the dissipative Josephson junction serves as an ideal condensed-matter example of open systems where the domain-wall solution, describing the position of the magnetic flux, appears as a sine-Gordon kink. The second purpose of this paper is to investigate general consequences of the one-dimensional translational symmetry breaking in open systems. In fact, it remains unclear what is the universal property of the fluctuating domain wall in general nonequilibrium open systems in sharp contrast to those in closed systems.

To accomplish twofold goals, we rely on the symmetry-based field-theoretical approach to the nonequilibrium dynamics, whose basis is recently developed in constructing the EFT for a dissipative fluid in closed systems [38–48] and generalized to describe the NG mode in open systems [49–53] (see also Ref. [54] for a holographic realization of the NG mode in open systems). In particular, we rely on the path-integral formalism from two different viewpoints—a bottom-up view from the Martin-Siggia-Rose (MSR) formalism for classical stochastic systems [55–57] and a top-down view from the Schwinger-Keldysh formalism for quantum open systems [58–63]. In both views, a recent perspective of the symmetry structure of open systems clarified in Refs. [49–53] is crucial. The notion of the symmetry becomes a little complicated since the corresponding physical charge in open systems is no longer conserved due to the dissipative coupling to the environment. Nevertheless, we can still define the spontaneous symmetry breaking and the associated NG mode in open systems [49–53]. The symmetry peculiar to open systems is different from the approximate symmetry that is explicitly but weakly broken. Accordingly, the behavior of NG modes discussed in this paper is also different from the gapped and overdamped pseudo-NG modes associated with approximate translational symmetry breaking [64–66].

In the first part (Sec. 2), we take a bottom-up route, starting from the classical stochastic description of the (2+1)-dimensional dissipative Josephson junction. Using the Fokker-Planck (operator) and the MSR (path-integral) formalisms, we clarify the symmetry structure peculiar to the open systems, and then derive the effective theory for the domain-wall fluctuation on the top of the sine-Gordon kink. The resulting energy spectrum of the fluctuation in two regimes—thin-wall and thick-wall regimes—shows the appearance of a diffusive pair mode; a gapless diffusion mode and its gapped partner. In the second part (Sec. 3-4), we take a top-down route, relying on the symmetry-based Schwinger-Keldysh formalism, which has been applied to the NG mode attached to the SSB for the time-translational and internal (or on-site) symmetries in open systems [51, 53]. Based on the symmetry and Schwinger-Keldysh requirements, we construct the most general effective Lagrangian for open systems with the one-dimensional translational symmetry breaking. The resulting effective Lagrangian demonstrates that the presence of a pair of the diffusive NG mode is a universal result in the thin-wall regime, while it also has a peculiar term possibly leading to a linear propagation and instability of the NG mode in the thick-wall regime. We also find that the same peculiar term leads to the nonlinear cubic

interaction term in the thin-wall regime, which may induce the KPZ universality class [25–32]. Our results show that the presence of the diffusive NG mode is a universal property of the stable domain wall in nonequilibrium open systems, and there is a model-dependent peculiar term that could induce the KPZ universality class.

The organization of the paper is in order: In Sec. 2, we investigate the fluctuation dynamics of the domain wall in the dissipative Josephson junction. In Sec. 3, we briefly review the Schwinger-Keldysh formalism in preparation to formulate the EFT for open systems. In Sec. 4, we construct the effective Lagrangian of open systems with translational symmetry breaking to investigate the universal property of the dissipative domain wall. Sec. 5 is devoted to the summary and outlook. In Appendix A, we present an EFT for the domain wall in finite-temperature closed systems in comparison to that given in the main text.

## 2 Dissipative domain wall in Josephson junction

In this section, we investigate the domain-wall dynamics in the dissipative Josephson junction with noise. While the subject of this section is interesting in its own right, it also illustrates our general motivation and formulation on spontaneous translational symmetry breaking and the resulting dynamics in open systems given in the subsequent sections. We provide the model describing the dissipative Josephson junction with noise in Sec. 2.1. To discuss the symmetry breaking of the stationary solution of the model, we introduce the operator and path integral formalisms in Sec. 2.2 and identify a peculiar symmetry structure in open systems in Sec. 2.3. In Sec. 2.4, we show that the stationary solution spontaneously breaks the translational symmetry peculiar to open systems and investigate the dynamics of the NG field associated with the symmetry breaking.

### 2.1 Dissipative sine-Gordon model with noise

The Josephson junction consists of two superconducting electrodes separated by a few nanometer thin layers of the insulator. By applying a uniform magnetic field parallel to the layer, a magnetic flux $\phi \in [0, 2\pi]$ sticks to the insulator (see the left panel of Fig. 1). The dynamics of the magnetic flux is described by the dissipative sine-Gordon equation [34, 35]:

$$\partial_t^2 \phi(t, \boldsymbol{x}) - \boldsymbol{\nabla}^2 \phi(t, \boldsymbol{x}) + m^2 \sin \phi(t, \boldsymbol{x}) + \alpha \partial_t \phi(t, \boldsymbol{x}) - \beta \boldsymbol{\nabla}^2 \partial_t \phi(t, \boldsymbol{x}) = \xi(t, \boldsymbol{x}), \quad (1)$$

where we scaled the space and time length to make the coefficients of the first two terms on the left-hand side to be unity. The mass term comes from the Josephson current due to the phase difference between the two superconducting electrodes, associated with the magnetic flux. Compared with the sine-Gordon model in closed systems, we have three additional terms $\alpha \partial_t \phi$, $\beta \boldsymbol{\nabla}^2 \partial_t \phi$ and $\xi$, describing effects of the dissipation and noise. The terms proportional to $\alpha$ and $\beta$ captures the dissipative effect from quasi-particle tunneling and the surface resistance of the superconductors, respectively, while $\xi$ corresponds to a bias current density [34, 35]. We also take account of the possible fluctuating property of the bias current density $\xi(t, \boldsymbol{x})$, whose stochastic property is assumed to be a Gaussian white noise with no bias [36, 37]:

$$\langle \xi(t, \boldsymbol{x}) \rangle_\xi = 0, \qquad \langle \xi(t, \boldsymbol{x}) \xi(t', \boldsymbol{x}') \rangle_\xi = A \delta(t - t') \delta^2(\boldsymbol{x} - \boldsymbol{x}'), \quad (2)$$

where $\langle \cdots \rangle_\xi$ represents the average over the noise $\xi$. The parameter $A$ characterizes the magnitude of the bias current noise. Although the noise magnitude $A$ is usually related to the friction magnitudes by the fluctuation-dissipation relation, we do not assume such relations to carry out a general analysis. In experimental realizations, these dissipative effects are inevitable at a finite temperature, and typically the noise magnitude $A$ is proportional to the

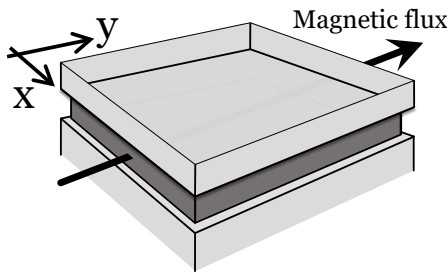
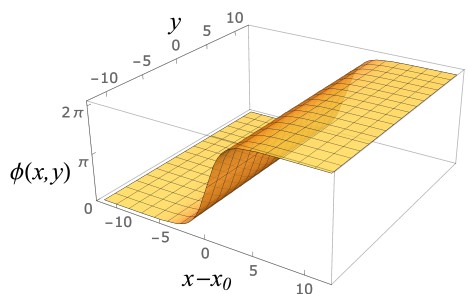

Figure 1: Left: A magnetic flux applied to a insulator layer by a uniform magnetic field in $y$-direction. Right: $2\pi$-kink phase difference $\phi(t, \boldsymbol{x})$ induced by the magnetic flux ($m = 1.0$).

temperature of the environment. Due to the terms proportional to $\alpha, \beta$ as well as the noise, Eq. (1) describes an open system exposed to the dissipation.

The vital point for our subsequent analysis is that Eq. (1) in the mean-field limit, where the right-hand side is replaced by its averaged value 0, supports the following sine-Gordon kink as the domain-wall solution:

$$\bar{\phi}(x - x_0) = 4 \arctan\left[e^{m(x-x_0)}\right], \tag{3}$$

where we imposed the following boundary condition

$$\lim_{x \to -\infty} \phi(t, x, y) = 0 \quad \text{and} \quad \lim_{x \to \infty} \phi(t, x, y) = 2\pi. \tag{4}$$

This solution describes a domain wall localized at position $x_0$ as shown in the right panel of Fig. 1. The imposed boundary condition (4) means that there is one magnetic flux line piercing the sandwiched insulator along the $y$-direction (see Fig. 1). Thus, the $(2 + 1)$-dimensional Josephson junction realizes the domain-wall solution (3) when we apply an appropriate amount of the magnetic field parallel to the layer to impose the boundary condition (4). In the following of this section, we will clarify the dynamics of the realized domain wall based on the symmetry of the dissipative Josephson junction.

## 2.2 Operator and path integral formalisms for stochastic dynamics

The crucial point for the domain-wall dynamics is that the presence of the wall breaks a spatial translational invariance. In closed systems, this spontaneous symmetry breaking results in a gapless collective excitation identified as the NG mode, which dominates the low-energy dynamics of the wall. However, since the dissipative sine-Gordon equation (1) describes the open system exposed to the dissipation and noise that break the momentum conservation, we need to be careful about what is the symmetry of our open system [49–53].

We shall now clarify the notion of the symmetry and the NG mode in open systems. For that purpose, it is useful to rely on operator and path-integral formalisms for the stochastic equation of motion, which are known as the Fokker-Planck formalism and the MSR formalism [55–57], respectively (see, e.g., Ref. [67] for a review). In particular, we mainly employ the MSR formalism, which directly leads to the effective Lagrangian of the NG mode associated with the domain wall.

In preparation for moving to the Fokker-Planck operator formalism, we first introduce a field variable $\chi(t, \boldsymbol{x})$ conjugate to $\phi(t, \boldsymbol{x})$ and decompose Eq. (1) as

$$\partial_t \phi(t, \boldsymbol{x}) = \chi(t, \boldsymbol{x}), \tag{5a}$$

$$\partial_t \chi(t, \boldsymbol{x}) = \boldsymbol{\nabla}^2 \phi(t, \boldsymbol{x}) - m^2 \sin\phi(t, \boldsymbol{x}) - \alpha\chi(t, \boldsymbol{x}) + \beta\boldsymbol{\nabla}^2\chi(t, \boldsymbol{x}) + \xi(t, \boldsymbol{x}). \tag{5b}$$

Then, we introduce the probability distribution for $\phi$ and $\chi$ as

$$\mathcal{P}[t;\phi_R(\boldsymbol{x}),\chi_R(\boldsymbol{x})] \equiv \prod_{\boldsymbol{x}} \langle \delta(\phi_R(\boldsymbol{x}) - \phi(t,\boldsymbol{x}))\delta(\chi_R(\boldsymbol{x}) - \chi(t,\boldsymbol{x}))\rangle_\xi. \tag{6}$$

Note that $\phi_R(\boldsymbol{x})$ and $\chi_R(\boldsymbol{x})$ denote c-number field configurations while $\phi(t,\boldsymbol{x})$ and $\chi(t,\boldsymbol{x})$ are solutions of the stochastic equations of motion (5). Note that $\phi_R(\boldsymbol{x})$ is also in $[0,2\pi]$ as well as the magnetic flux $\phi(t,\boldsymbol{x})$. Equation (6) defines the probability distribution functional that the field variables $\{\phi(t,\boldsymbol{x}),\chi(t,\boldsymbol{x})\}$ realize a configuration $\{\phi_R(\boldsymbol{x}),\chi_R(\boldsymbol{x})\}$ at a given time $t$.

Using the equations of motion (5), we can show that the time evolution of the probability distribution $\mathcal{P}[t;\phi_R(\boldsymbol{x}),\chi_R(\boldsymbol{x})]$ is described by the Fokker-Planck equation

$$\partial_t \mathcal{P}[t;\phi_R(\boldsymbol{x}),\chi_R(\boldsymbol{x})] = -H_{\mathrm{FP}}\mathcal{P}[t;\phi_R(\boldsymbol{x}),\chi_R(\boldsymbol{x})], \tag{7}$$

where we introduced the Fokker-Planck Hamiltonian as

$$
\begin{aligned}
H_{\mathrm{FP}} \equiv \int \mathrm{d}^2x \Bigg[ &\frac{\delta}{\delta\phi_R(\boldsymbol{x})}\chi_R(\boldsymbol{x}) \\
&+ \frac{\delta}{\delta\chi_R(\boldsymbol{x})}\Big[\boldsymbol{\nabla}^2\phi_R(\boldsymbol{x}) - m^2\sin\phi_R(\boldsymbol{x}) - \alpha\chi_R(\boldsymbol{x}) + \beta\boldsymbol{\nabla}^2\chi_R(\boldsymbol{x})\Big] - \frac{A}{2}\frac{\delta^2}{\delta\chi_R(\boldsymbol{x})^2}\Bigg].
\end{aligned}
\tag{8}
$$

Notice that the Fokker-Planck equation (7) looks similar to the Schrödinger equation for the wave function in quantum theory. Motivated by this observation, we define *field operators* by

$$\hat{\phi}_R(\boldsymbol{x}) = \phi_R(\boldsymbol{x}), \quad \hat{\chi}_R(\boldsymbol{x}) = \chi_R(\boldsymbol{x}), \quad \hat{\chi}_A(\boldsymbol{x}) = -\mathrm{i}\frac{\delta}{\delta\phi_R(\boldsymbol{x})}, \quad \hat{\phi}_A(\boldsymbol{x}) = +\mathrm{i}\frac{\delta}{\delta\chi_R(\boldsymbol{x})}. \tag{9}$$

Introducing the commutation relation as $[\hat{A},\hat{B}] = \hat{A}\hat{B} - \hat{B}\hat{A}$, one finds that the above operators, by definition, satisfy the canonical commutation relations

$$[\hat{\phi}_R(\boldsymbol{x}), \hat{\chi}_A(\boldsymbol{x}')] = [\hat{\phi}_A(\boldsymbol{x}), \hat{\chi}_R(\boldsymbol{x}')] = \delta^{(2)}(\boldsymbol{x} - \boldsymbol{x}'), \tag{10}$$

where the other commutators vanish. Therefore, we can regard the Fokker-Planck equation (7) as the analogue of the Schrödinger equation with imaginary time.

On the other hand, it should be also emphasized remarkable differences between the Fokker-Planck equation and the ordinary Schrödinger equation. First, $\mathcal{P}[t;\phi_R(\boldsymbol{x}),\chi_R(\boldsymbol{x})]$ in the Fokker-Planck formalism describes the real-valued probability distribution while the wave function in quantum theory does the complex-valued function, whose square gives the probability distribution. Second, the Fokker-Planck Hamiltonian is not the Hermitian operator in sharp contrast to the usual Hamiltonian in quantum theory. As a result, despite the similar structure with quantum theory, the low-energy spectrum of the resulting NG mode will be qualitatively different.

As in quantum theory, instead of the operator formalism, we can use the equivalent path-integral (or Lagrangian) formalism known as the MSR formalism [55–57]. In fact, we can perform a systematic computation of the correlation function based on the path-integral formula for the generating functional $Z[j_\phi, j_\chi]$ given by

$$
\begin{aligned}
Z[j_\phi, j_\chi] &\equiv \langle \mathrm{e}^{\mathrm{i}\int \mathrm{d}t\,\mathrm{d}^2x\,[j_\phi(t,\boldsymbol{x})\phi(t,\boldsymbol{x}) + j_\chi(t,\boldsymbol{x})\chi(t,\boldsymbol{x})]}\rangle_\xi \\
&= \int \mathcal{D}\phi_R\mathcal{D}\chi_R\mathcal{D}\phi_A\mathcal{D}\chi_A\, \mathrm{e}^{\mathrm{i}S_{\mathrm{MSR}}[\phi_R,\chi_R,\phi_A,\chi_A] + \mathrm{i}\int \mathrm{d}t\,\mathrm{d}^2x\,[j_\phi(t,\boldsymbol{x})\phi_R(t,\boldsymbol{x}) + j_\chi(t,\boldsymbol{x})\chi_R(t,\boldsymbol{x})]},
\end{aligned}
\tag{11}
$$

where $j_\phi$ and $j_\chi$ are the source fields to compute correlation functions of stochastic variables $\phi(t,\boldsymbol{x})$ and $\chi(t,\boldsymbol{x})$. In the second line, we introduced the auxiliary fields $\phi_A$ and $\chi_A$ to make

$\phi_R$ and $\chi_R$ as solutions of the equations of motion (5) and performed the integration over the noise. We also dropped a Jacobian factor since it does not play an important role in the following analysis. The phase space MSR action $iS_{\text{MSR}}$ is given by

$$iS_{\text{MSR}} = \int dt d^2x \Big[ i\chi_A \partial_t \phi_R - i\phi_A \partial_t \chi_R - H_{\text{FP}} \Big]$$

$$= \int dt d^2x \Big[ i\chi_A \big(\partial_t \phi_R - \chi_R\big) - i\phi_A \big(\partial_t \chi_R - \nabla^2 \phi_R + m^2 \sin\phi_R + \alpha\chi_R - \beta\nabla^2\chi_R\big) - \frac{A}{2}\phi_A^2 \Big]. \tag{12}$$

Furthermore, integrating out the conjugate variables $\chi_R$ and $\chi_A$, one can also find the configuration space MSR action for the dissipative sine-Gordon model as follows:

$$iS_{\text{MSR}}[\phi_R, \phi_A] = \int dt d^2x \Big[ -i\phi_A \big(\partial_t^2 \phi_R - \nabla^2 \phi_R + m^2 \sin\phi_R + \alpha\partial_t\phi_R - \beta\nabla^2\partial_t\phi_R\big) - \frac{A}{2}\phi_A^2 \Big]. \tag{13}$$

Note that $\phi_R$ corresponds to the physical quantity describing the original magnetic flux, whereas $\phi_A$ is an auxiliary field to make $\phi_R$ as a solution of the original Langevin equation. In the noiseless limit $A \to 0$, this action reduces to the Fourier expression of the delta functional using the auxiliary field $\phi_A$, which restricts field configurations of $\phi_R$ to be those satisfying the deterministic dissipative sine-Gordon equation. In other words, the last term in Eq. (13) captures the effect of the noise. Likewise, in the phase space action, $\chi_R$ is the physical quantity while $\chi_A$ is an auxiliary field.

## 2.3 Translational symmetry

Based on the Fokker-Planck and MSR formalisms presented in the previous section, we discuss spatial translational symmetries in the open system. In the MSR formalism, we define the symmetry as the invariance of the MSR action (12) or (13) under the corresponding transformation. The equivalent definition of the symmetry in the Fokker-Planck formalism is given by the charge operator commuting with the Fokker-Planck Hamiltonian (recall that the Fokker-Planck Hamiltonian generates the time translation).

We then investigate the symmetry of the dissipative Josephson junction. First of all, note that the MSR action Eq. (12) and Eq. (13) does not have an explicit coordinate dependence: namely, our model is invariant under the following transformation:

$$\begin{cases} \phi_R(t, \mathbf{x}) \to \phi_R'(t, \mathbf{x}) = \phi_R(t, \mathbf{x} + \boldsymbol{\epsilon}_A), \\ \phi_A(t, \mathbf{x}) \to \phi_A'(t, \mathbf{x}) = \phi_A(t, \mathbf{x} + \boldsymbol{\epsilon}_A), \\ \chi_R(t, \mathbf{x}) \to \chi_R'(t, \mathbf{x}) = \chi_R(t, \mathbf{x} + \boldsymbol{\epsilon}_A), \\ \chi_A(t, \mathbf{x}) \to \chi_A'(t, \mathbf{x}) = \chi_A(t, \mathbf{x} + \boldsymbol{\epsilon}_A), \end{cases} \tag{14}$$

associated with the spatial translation $\mathbf{x} \to \mathbf{x}' = \mathbf{x} - \boldsymbol{\epsilon}_A$. The corresponding conserved Noether charge is provided by

$$P_{i,A} \equiv \int d^2x \Big[ \chi_R(t, \mathbf{x}) \partial_i \phi_A(t, \mathbf{x}) + \chi_A(t, \mathbf{x}) \partial_i \phi_R(t, \mathbf{x}) \Big], \tag{15}$$

with $\partial_i = (\partial_x, \partial_y)$. In the Fokker-Planck formalism, the operator version of this Noether charge generates the spatial translation. In fact, by the use of the commutation relation (10), one can reproduce the infinitesimal transformation rule of, e.g., $\phi_R$ in Eq. (14) as

$$\delta_{\boldsymbol{\epsilon}_A} \hat{\phi}_R(\mathbf{x}) \equiv i \big[ \boldsymbol{\epsilon}_A \cdot \hat{\mathbf{P}}_A, \hat{\phi}_R(\mathbf{x}) \big] = \boldsymbol{\epsilon}_A \cdot \nabla \hat{\phi}_R(\mathbf{x}), \tag{16}$$

and others follow in the same way. One can also check the spatial translational symmetry as the commutativity of the Noether charge $\hat{P}_{i,A}$ with the Fokker-Planck Hamiltonian $[\hat{H}_{\mathrm{FP}}, \hat{P}_{i,A}] = 0$. We refer to this symmetry as $P_{i,A}$-symmetry.

Although the operator $\hat{P}_{i,A}$ gives a conserved charge generating the spatial translation (14), this quantity does not represent a physical momentum because $\hat{P}_{i,A}$ involves the auxiliary fields $\phi_A$ and $\chi_A$. As in the non-dissipative sine-Gordon model, the physical momentum should be defined solely by the physical fields $\phi_R$ and $\chi_R$ as

$$\hat{P}_{i,R} \equiv \int \mathrm{d}^2 x \, \hat{\chi}_R(\boldsymbol{x}) \partial_i \hat{\phi}_R(\boldsymbol{x}). \tag{17}$$

In contrast to $\hat{P}_{i,A}$, the physical momentum $\hat{P}_{i,R}$, however, is not conserved because it does not commute with $\hat{H}_{\mathrm{FP}}$ as

$$\mathrm{i}[\hat{H}_{\mathrm{FP}}, \hat{P}_{i,R}] = \int \mathrm{d}^2 x \left[ \left( -\alpha \hat{\chi}_R + \beta \boldsymbol{\nabla}^2 \hat{\chi}_R + \mathrm{i} A \hat{\phi}_A \right) \partial_i \hat{\phi}_R \right] + \text{(surface term)}. \tag{18}$$

The non-vanishing contribution results from the terms proportional to $\alpha$, $\beta$, and $A$, and thus, the presence of the dissipation and noise makes the physical momentum $\hat{P}_{i,R}$ to be nonconserved. This is because the physical momentum accompanied by the magnetic flux $\phi_R$ diffuses into the environment and becomes a non-conserved quantity in the open system. In other words, there is no $P_{i,R}$-symmetry in contrast to closed systems.

The above structure is a salient feature of open systems: the physical charge $\hat{P}_{i,R}$ is not conserved due to the dissipation and noise while there is a $P_{i,A}$-symmetry generated by the conserved auxiliary charge $\hat{P}_{i,A}$. The crucial point here is that it is possible for a steady-state solution to spontaneously break the present $P_{i,A}$-symmetry. Following the definition in closed systems, we define the spontaneous $P_{i,A}$-symmetry breaking in the dissipative Josephson junction systems by the existence of a certain physical order parameter field $\Phi_R(t, \boldsymbol{x})$ as follows:

$$\exists \, \Phi_R(t, \boldsymbol{x}) \quad \text{such that} \quad \langle \delta_{\epsilon_A} \Phi_R(t, \boldsymbol{x}) \rangle = \langle [\mathrm{i}\epsilon_A \cdot \hat{\boldsymbol{P}}_A, \hat{\Phi}_R(t, \boldsymbol{x})] \rangle \neq 0, \tag{19}$$

where $\langle \cdots \rangle$ denotes the path-integral average with the MSR action (13). One can find that the condition for the order parameter field is simply given by $\partial_i \langle \Phi_R(t, \boldsymbol{x}) \rangle \neq 0$, so that it precisely indicates the inhomogeneity of the steady-state solution. Since the mean-field solution (3) indeed breaks the translational symmetry associated with the conserved charge $\hat{P}_{x,A}$, we can investigate the domain-wall dynamics in open systems from the perspective of the symmetry breaking. Although the origin of the two kinds of charges and symmetry structures may be unclear so far, we will see that they naturally arise from the underlying quantum theory based on the Schwinger-Keldysh formalism in Sec. 3.

## 2.4 Effective Lagrangian for the domain-wall fluctuation

Let us investigate the domain-wall dynamics of the dissipative sine-Gordon model using the configuration space MSR action (13). First of all, the equations of motion in the MSR formalism are given by

$$0 = \frac{\delta S_{\mathrm{MSR}}[\phi_R, \phi_A]}{\delta \phi_A(t, \boldsymbol{x})} = -\partial_t^2 \phi_R + \boldsymbol{\nabla}^2 \phi_R - m^2 \sin \phi_R - \alpha \partial_t \phi_R + \beta \boldsymbol{\nabla}^2 \partial_t \phi_R + \mathrm{i} A \phi_A, \tag{20a}$$

$$0 = \frac{\delta S_{\mathrm{MSR}}[\phi_R, \phi_A]}{\delta \phi_R(t, \boldsymbol{x})} = -\partial_t^2 \phi_A + \boldsymbol{\nabla}^2 \phi_A - m^2 \phi_A \cos \phi_R + \alpha \partial_t \phi_A - \beta \boldsymbol{\nabla}^2 \partial_t \phi_A. \tag{20b}$$

To solve these equations, we employ a mean-field approximation in which the solution satisfies $S_{\mathrm{MSR}}[\phi_R, \phi_A] = 0$. This condition comes from the fact that the generating functional

without the external fields satisfies $Z = 1$ because of Eq. (11). With the use of Eq. (20a), the MSR action turns into $S_{\text{MSR}}[\phi_R, \phi_A] = \int dt d^2 x \, A \phi_A^2/2$, which leads to $\phi_A = 0$ in the mean-field approximation. As is expected, with the boundary condition $\lim_{x \to -\infty} \phi_R(t, \boldsymbol{x}) = 0$ and $\lim_{x \to \infty} \phi_R(t, \boldsymbol{x}) = 2\pi$ corresponding to Eq. (4), one finds the stationary domain-wall solution in the mean-field approximation, given by

$$\phi_R(t, \boldsymbol{x}) = \bar{\phi}(x - x_0) = 4 \arctan(e^{m(x-x_0)}), \qquad \phi_A(t, \boldsymbol{x}) = 0. \tag{21}$$

Thus, identifying the order parameter field as $\phi_R(t, \boldsymbol{x})$, we find that the domain-wall solution spontaneously breaks $P_{x,A}$-symmetry. From the path-integral viewpoint, this solution gives a saddle-point solution describing the domain wall.

We then consider the fluctuation on the top of the saddle point domain-wall solution (21). To parametrize the fluctuation around the realized domain wall, we introduce field variables $\pi_R(t, \boldsymbol{x})$ by promoting $x_0$ to a dynamical field, and $\pi_A(t, \boldsymbol{x})$ as follows:

$$\phi_R(t, \boldsymbol{x}) = \bar{\phi}(x + \pi_R(t, \boldsymbol{x})), \qquad \phi_A(t, \boldsymbol{x}) = \bar{\phi}'(x + \pi_R(t, \boldsymbol{x}))\pi_A(t, \boldsymbol{x}), \tag{22}$$

with $\bar{\phi}'(x) \equiv \partial_x \bar{\phi}(x)$. One can regard Eq. (22) as a field redefinition useful to analyze the fluctuation around the domain-wall solution located at $x = 0$. As we will see later, the fluctuation fields $\pi_R$ and $\pi_A$ include a gapless mode, so that we refer to these fields as NG fields. We put a little complicated prefactor of $\pi_A$ to assign it to the same dimension as $\pi_R$. The reason for this choice will be clarified from the underlying Schwinger-Keldysh viewpoint around Eq. (67). We also note that $\bar{\phi}(x)$ satisfies

$$\bar{\phi}''(x) = m^2 \sin \bar{\phi}(x). \tag{23}$$

Substituting the parametrization (22) into the original MSR action (13), we can derive the effective action for the fluctuation ($\pi_R, \pi_A$) as

$$iS_{\text{MSR}}[\pi_R, \pi_A] = i \int dt d^2 x \big[\mathcal{L}^{(2)} + \mathcal{L}^{(\text{int})}\big]. \tag{24}$$

Here, we introduced the quadratic (interacting) part of the effective Lagrangian as $\mathcal{L}^{(2)}$ ($\mathcal{L}^{(\text{int})}$) based on the expansion with respect to the fluctuation fields $\pi_R$ and $\pi_A$. From the direct computation with the help of Eq. (23), we obtain the quadratic part as

$$\begin{aligned}
\mathcal{L}^{(2)} = &-\bar{\phi}'(x)^2 \pi_A \big[\partial_t^2 + \alpha \partial_t - \boldsymbol{\nabla}^2 - \beta \partial_t \boldsymbol{\nabla}^2\big]\pi_R \\
&+ 2\bar{\phi}'(x)\bar{\phi}''(x)\pi_A(\partial_x \pi_R + \beta \partial_t \partial_x \pi_R) + \beta \bar{\phi}'(x)\bar{\phi}'''(x)\pi_A \partial_t \pi_R + \frac{iA}{2}\bar{\phi}'(x)^2 \pi_A^2.
\end{aligned} \tag{25}$$

One may think that this quadratic part $\mathcal{L}^{(2)}$ looks more complicated than the original MSR action. In fact, even if we neglect the interaction terms, the coefficients appearing in $\mathcal{L}^{(2)}$ are $x$-dependent, which makes the further analysis difficult. This difficulty results from the fact that the effective Lagrangian (25) still keeps a bunch of gapped excitations [recall that Eq. (25) is obtained just by the field redefinition without focusing on the low-energy regime]. However, it is possible to drastically simplify the analysis by focusing only on the gapless mode. In the following, we consider two different regimes, in which such simplification is available; that is, the thin-wall and the thick-wall regimes.

### 2.4.1 Low-energy spectrum in thin-wall regime

We first consider the thin-wall regime, which corresponds to the usual low-energy limit of the domain-wall dynamics. One can regard this regime as the case where the length scale of the domain-wall fluctuation of our interest is sufficiently larger than the thickness of the wall.

Let us then investigate the low-energy spectrum for the domain-wall fluctuation. The vital point here is that $x$-dependent coefficients such as $\bar{\phi}'(x)^2$ take the nonvanishing value only near the domain-wall position $x = 0$. For this reason, we rely on the ansatz that the NG fields $\pi_R$ and $\pi_A$ are localized at the domain-wall position $x = 0$ as $\widetilde{\pi}_{R/A}(t, y) \equiv \pi_{R/A}(t, x = 0, y)$ (see, e.g., Ref. [22]). This ansatz enables us to simplify the action (25) to the low-energy MSR effective action for the NG fields $\widetilde{\pi}_R$ and $\widetilde{\pi}_A$ as

$$
\begin{aligned}
\mathrm{i}S_{\text{thin}} &= \int \mathrm{d}t\mathrm{d}y\, 8m\left[-\mathrm{i}\widetilde{\pi}_A(t, y)\left(\partial_t^2 + \gamma \partial_t - \beta \partial_t \partial_y^2 - \partial_y^2\right)\widetilde{\pi}_R(t, y) - \frac{A}{2}\widetilde{\pi}_A(t, y)^2 + O(\pi^4)\right] \\
&= -\frac{1}{2}\int \mathrm{d}t\mathrm{d}y \begin{pmatrix}\widetilde{\pi}_R(t, y) & \widetilde{\pi}_A(t, y)\end{pmatrix}\begin{pmatrix}0 & \mathrm{i}G_{A;\perp}^{-1} \\ \mathrm{i}G_{R;\perp}^{-1} & 8mA\end{pmatrix}\begin{pmatrix}\widetilde{\pi}_R(t, y) \\ \widetilde{\pi}_A(t, y)\end{pmatrix} + O(\pi^4),
\end{aligned}
\tag{26}
$$

with the effective damping constant $\gamma \equiv \alpha + m^2\beta/3$. In the second line, we introduced the inverse of the retarded/advanced Green's function $G_{R/A;\perp}^{-1}$ as

$$
G_{R;\perp}^{-1} = 8m[\partial_t^2 + \gamma \partial_t - \beta \partial_t \partial_y^2 - \partial_y^2] \quad \text{and} \quad G_{A;\perp}^{-1} = 8m[\partial_t^2 - \gamma \partial_t + \beta \partial_t \partial_y^2 - \partial_y^2].
\tag{27}
$$

Besides, from the lower-right component of the matrix in Eq. (26), we can also find the symmetric Green's function $G_{RR;\perp}$, which describes the correlation function of the original stochastic variables $\phi$. In the Fourier space, it is given by

$$
G_{RR;\perp}(\omega, k_y) = 8mAG_{R;\perp}(\omega, k_y)G_{A;\perp}(\omega, k_y) = \frac{A}{8m}\frac{1}{(\omega^2 - k_y^2)^2 + \omega^2(\gamma + \beta k_y^2)^2}.
\tag{28}
$$

We note that the cubic interaction term, corresponding to the nonlinear term in the KPZ equation, disappears in the thin-wall regime. We will revisit this point in Sec. 4.6 from a general EFT viewpoint.

The above result enables us to clarify the low-energy spectrum of the NG fields $\widetilde{\pi}_R$ and $\widetilde{\pi}_A$. We can find the pole location of the retarded Green's function by solving

$$
0 = G_{R;\perp}^{-1}(\omega, k_y) = 8m(-\omega^2 - \mathrm{i}\gamma\omega - \mathrm{i}\beta\omega k_y^2 + k_y^2).
\tag{29}
$$

This allows us to identify the dispersion relation of the NG fields $\widetilde{\pi}_R$ and $\widetilde{\pi}_A$ as

$$
\omega(k_y) = -\mathrm{i}\frac{\gamma + \beta k_y^2}{2} \pm \mathrm{i}\sqrt{\left(\frac{\gamma + \beta k_y^2}{2}\right)^2 - k_y^2} = \begin{cases} -\dfrac{\mathrm{i}}{\gamma}k_y^2 + O(k_y^4), \\ -\mathrm{i}\gamma - \mathrm{i}(\beta - \gamma^{-1})k_y^2 + O(k_y^4). \end{cases}
\tag{30}
$$

Figure 2 shows the dispersion relation and the symmetric Green's function for $\widetilde{\pi}_R$ and $\widetilde{\pi}_A$. We clearly see that $\widetilde{\pi}_R$ and $\widetilde{\pi}_A$ describe a pair of the gapless mode (NG mode) and its gapped partner. In sharp contrast to the NG mode in closed systems, the dispersion relation only has the negative imaginary part in the low-wavenumber limit. The absence of the real part indicates that the NG mode associated with the domain wall in open systems diffuses without propagation. This is a salient feature of the NG mode in open systems [49–53].[1]

One can also confirm the consistency with the usual propagating NG mode in the closed system. In fact, when we consider the smaller value for the dissipative couplings $\alpha$ and $\beta$, the relaxational gap for the gapped partner becomes smaller. At the vanishing the dissipative coupling with $\alpha = \beta = 0$, Eq. (29) eventually reproduces the gapless linear dispersion relation for (a pair of) the propagating NG mode. This is consistent with the result for the zero-temperature domain wall in closed systems (see, e.g., Ref. [21] for a symmetry-based approach for the domain-wall dynamics at zero temperature).

---

[1]The propagator (29) has the same form as that of the telegraphic equation. Therefore, the similar spectrum has been discussed in various systems (see, e.g., Refs. [68–70] for recent discussions on the transverse wave in liquids).

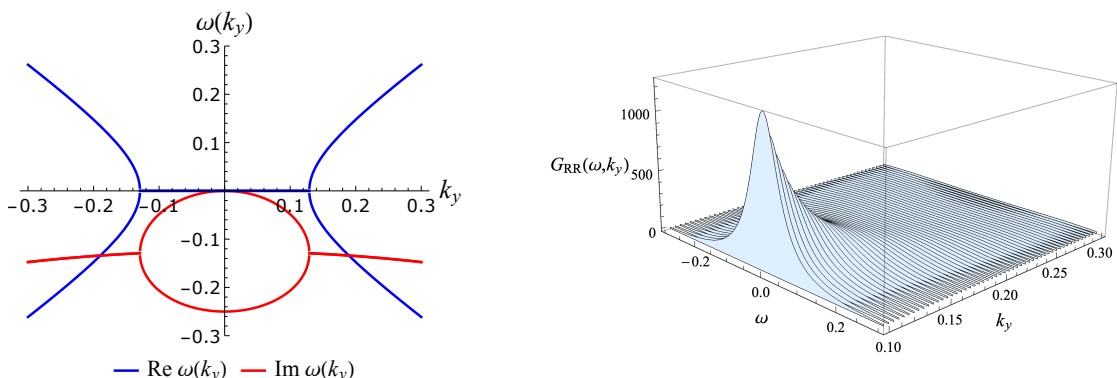

Figure 2: The dispersion relation (30) (left) and the symmetric Green's function (28) (right) of the NG mode in the thin-wall regime with $(\gamma, \beta, A, m) = (0.25, 0.5, 1.0, 1.0)$.

### 2.4.2 Low-energy spectrum in thick-wall regime

Let us next consider the opposite of the thin-wall regime, called the thick-wall regime. In this regime, we consider the dynamics of the fluctuation carrying momentum larger than $m$. In other words, for fluctuations under consideration, the wall thickness $m^{-1}$ is sufficiently large, and they feel as if there is a constant slope continuing endlessly. This is a spatial analogue of the slow-roll inflation in cosmology [71,72].

One can take the thick-wall regime by approximating the domain-wall configuration as

$$\bar{\phi}'(x) = \frac{4m e^{mx}}{1 + e^{2mx}} \simeq 2m. \tag{31}$$

Then, we can drastically simplify the effective Lagrangian (25) by setting all higher derivatives like $\bar{\phi}''(x)$ to zero. The resulting effective action in the thick-wall regime is

$$iS_{\text{thick}} = \int dt d^2x \, 4m^2 \Big[ -i\pi_A(t, \boldsymbol{x})(\partial_t^2 + \alpha\partial_t - \beta\partial_t\boldsymbol{\nabla}^2 - \boldsymbol{\nabla}^2)\pi_R(t, \boldsymbol{x}) - \frac{A}{2}\pi_A(t, \boldsymbol{x})^2 + O(\pi^4) \Big]. \tag{32}$$

Note that $\pi_R(t, \boldsymbol{x})$ and $\pi_A(t, \boldsymbol{x})$ in the thick-wall regime have a dependence on the spatial coordinate $x$ in contrast to that in the thin-wall one. From the effective action, we can read off a set of the Green's function and the dispersion relation as before. For instance, one finds the retarded Green's function in the Fourier space as

$$G_R^{-1}(\omega, \boldsymbol{k}) = 4m^2(-\omega^2 - i\alpha\omega - \beta\omega\boldsymbol{k}^2 + \boldsymbol{k}^2), \tag{33}$$

from which we can identify the dispersion relation of the NG mode as

$$\omega(\boldsymbol{k}) = -i\frac{\alpha + \beta\boldsymbol{k}^2}{2} \pm i\sqrt{\left(\frac{\alpha + \beta\boldsymbol{k}^2}{2}\right)^2 - \boldsymbol{k}^2} = \begin{cases} -\dfrac{i}{\alpha}\boldsymbol{k}^2 + O(\boldsymbol{k}^4), \\ -i\alpha - i(\beta - \alpha^{-1})\boldsymbol{k}^2 + O(\boldsymbol{k}^4), \end{cases} \tag{34}$$

with $\boldsymbol{k} = (k_x, k_y)$. Therefore, even in the thick-wall regime, there appear a gapless NG mode and its gapped partner as in the thin-wall one. We see that the translational symmetry along the $x$-direction effectively recovers for the NG field in the thick-wall regime, and furthermore, the dispersion relation in the present case becomes isotropic including the wavenumber $k_x$. We, however, note that the configuration of the original phase variable breaks the translational symmetry even in the thick-wall regime.

### 2.5 Underdamped Langevin dynamics in Josephson transmission line

Before closing this section, we investigate the dynamics of the kink in the Josephson transmission line (JTL), which is a one-dimensional Josephson junction system obtained after a dimensional reduction along $y$-direction (recall the left panel of Fig. 1). A magnetic flux stuck (or kink) in the JTL is often referred to as a "fluxon," and its dynamics has been studied from the viewpoint of the soliton [34–36]. In the present setup, the particle-like behavior of the fluxon is understood as a localized field configuration described by the dimensionally reduced thin-wall effective action

$$\mathrm{i}S_{\text{JTL}} = \int \mathrm{d}t\, 8m \left[ -\mathrm{i}q_A(t)\big(\partial_t^2 + \gamma \partial_t\big)q_R(t) - \frac{A}{2}q_A(t)^2 + O(q^4) \right], \tag{35}$$

where we introduced the localized position of the fluxon $q_{R/A}(t)$ by the dimensional reduction of the NG fields along the $y$-direction as $q_{R/A}(t) \equiv \tilde{\pi}_{R/A}(t, y = 0)$.

To investigate the fluxon dynamics, we apply a time-dependent external electric current to the JTL. Recalling that $\xi(t, \boldsymbol{x})$ in Eq. (1) describes the bias current density, we find the effect of the non-vanishing averaged current is captured by adding the following term in the original MSR action (13) (without the $y$-direction due to the reduction):

$$\mathrm{i}S_{\text{ext}} = \mathrm{i} \int \mathrm{d}t\mathrm{d}x\, \phi_A(t, x)J(t), \tag{36}$$

where $J(t)$ is a normalized bias current. To apply the analysis for the fluctuations around the steady state so far, we assume that the applied external current $J(t)$ is sufficiently small so as to perturb the position of the domain wall without collapsing it. After the same procedure to derive the low-energy effective action, this term is expressed in terms of $q_R(t)$ and $q_A(t)$ as

$$\mathrm{i}S_{\text{ext}} = \mathrm{i} \int \mathrm{d}t\mathrm{d}x\, \left[ \bar{\phi}'(x)J(t)q_A(t) + \bar{\phi}''(x)J(t)q_A(t)q_R(t) + O(q^3) \right]$$

$$= \mathrm{i} \int \mathrm{d}t\, \left[ 2\pi J(t)q_A(t) + O(q^3) \right], \tag{37}$$

where we used the thin-wall ansatz to obtain the second line. Thus, the fluxon dynamics driven by the external current is described by

$$\mathrm{i}S_{\text{JTL+ext}} = -\mathrm{i} \int \mathrm{d}t\, \left\{ q_A(t)\left[ 8m\big(\partial_t^2 + \gamma \partial_t\big)q_R(t) - 2\pi J(t) \right] - 4\mathrm{i}mAq_A(t)^2 + O(q^3) \right\}. \tag{38}$$

An intuitive understanding of the real-time fluxon dynamics is possible by translating back the MSR action (38) into the stochastic equation of motion. This translation is also useful to solve the initial value problem for the fluxon dynamics. Indeed, the quadratic action in Eq. (38) is identical to the MSR action of the simple Brownian motion driven by the external force [73]. As a result, the corresponding stochastic equation of motion for the fluxon is given by the underdamped Langevin equation:

$$\big(\partial_t^2 + \gamma \partial_t\big)q(t) = \frac{\pi}{4m}J(t) + \xi(t), \tag{39}$$

with the Gaussian white noise $\xi(t)$

$$\langle \xi(t) \rangle_\xi = 0, \qquad \langle \xi(t)\xi(t') \rangle_\xi = \frac{A}{8m}\delta(t - t'). \tag{40}$$

Suppose that we apply the external current $J(t)$ at time $t > 0$ with the initial condition for the fluxon as $q(0) = \partial_t q(0) = 0$. Then, one immediately find the solution of the Langevin equation as

$$q(t) = \bar{q}(t) + \frac{1}{\gamma}\int_0^t dt'\left(1 - e^{-\gamma(t-t')}\right)\xi(t') \quad \text{with} \quad \bar{q}(t) = \frac{\pi}{4m\gamma}\int_0^t dt'\left(1 - e^{-\gamma(t-t')}\right)J(t'). \tag{41}$$

Note that $\bar{q}(t)$ coincides with the averaged position as $\bar{q}(t) = \langle q(t)\rangle_\xi$, and thus, it is not a stochastic variable.

We shall then discuss experimental observables resulting from the fluxon dynamics. For that purpose, we recall the following relations between the phase variable $\phi(t,x)$ and the Josephson current $I(t,x)$ and voltage $V(t,x)$ (see, e.g., Ref. [6]):

$$I(t,x) = I_c \sin\phi(t,x), \qquad V(t,x) = \frac{1}{2e}\partial_t\phi(t,x), \tag{42}$$

where $I_c$ is the critical current of the Josephson junction and $e$ the elementary charge. The relation between the phase variable and the NG field (position variable) in the Langevin picture is also given by $\phi(t,x) = \bar{\phi}(x + q(t))$; the same one for the $R$-type variables [recall Eq. (22)]. By expanding this relation on the top of the averaged motion, we can express the experimental observables $I(t,x)$ and $V(t,x)$ in terms of the fluxon position as

$$I(t,x) \simeq I_c\left[\sin\bar{\phi}(x + \bar{q}(t)) + \delta q(t)\bar{\phi}'(x + \bar{q}(t))\cos\bar{\phi}(x + \bar{q}(t))\right], \tag{43}$$

$$V(t,x) \simeq \frac{1}{2e}\left[\bar{\phi}'(x + \bar{q}(t))\partial_t q(t) + \bar{\phi}''(x + \bar{q}(t))\delta q(t)\partial_t\bar{q}(t)\right], \tag{44}$$

where we introduced $\delta q(t) \equiv q(t) - \bar{q}(t)$ and neglected the higher-order $O(\delta q^2)$ terms.

With the help of Eq. (41), we can compute correlation functions for the Josephson current $I(t,x)$ and voltage $V(t,x)$. For example, we find the averaged current and voltage as

$$\langle I(t,x)\rangle_\xi \simeq I_c\sin\bar{\phi}(x + \bar{q}(t)), \quad \langle V(t,x)\rangle_\xi \simeq \frac{1}{2e}\bar{\phi}'(x + \bar{q}(t))\partial_t\bar{q}(t), \tag{45}$$

and their mean square variances at a large time $t \gg \gamma^{-1}$ as

$$\langle\delta I(t,x_1)\delta I(t,x_2)\rangle_\xi \sim \frac{AI_c^2 t}{8m\gamma^2}\bar{\phi}'(x_1 + \bar{q}(t))\bar{\phi}'(x_2 + \bar{q}(t))\cos\bar{\phi}(x_1 + \bar{q}(t))\cos\bar{\phi}(x_2 + \bar{q}(t)),$$

$$\langle\delta V(t,x_1)\delta V(t,x_2)\rangle_\xi \sim \frac{At}{32e^2 m\gamma^2}[\partial_t\bar{q}(t)]^2\bar{\phi}''(x_1 + \bar{q}(t))\bar{\phi}''(x_2 + \bar{q}(t)), \tag{46}$$

where we introduced the deviation of the Josephson current and voltage from their average values as

$$\delta I(t,x) \equiv I(t,x) - \langle I(t,x)\rangle_\xi, \qquad \delta V(t,x) \equiv V(t,x) - \langle V(t,x)\rangle_\xi. \tag{47}$$

The overall time-linear dependence of the variances in Eq. (46) is a manifestation of the Brownian motion of the fluxon.

The above results enable us to predict the spatiotemporal profile of the Josephson current and voltage, which can be measured in experiments by using a few parameters (the low-energy coefficients) $m$, $\gamma$, and $A$. For example, Fig. 3 demonstrates the fluxon position and the resulting voltage at a fixed position driven by the constant bias current $J(t) = \text{const}$. Since the fluxon motion induce the voltage localized at its position, the voltage takes a nonvanishing value when the fluxon passes through the position at which the measurement is performed [74, 75].

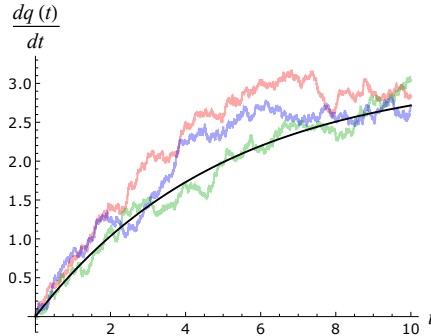
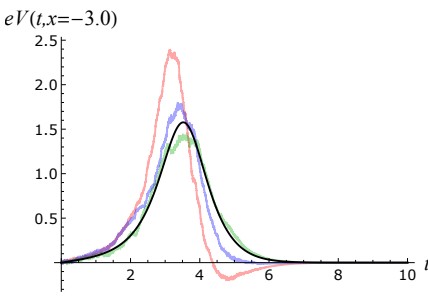

Figure 3: Solutions of the Langevin equation (39) (left panel) and the voltage at a given position $x = -3.0$ (right panel) a constant bias current with a parameter set $(m, \gamma, J, A) = (1.0, 0.2, 0.8, 0.5)$. Black lines shows the averaged result while the colored (red, green, and blue) ones are three sample solutions.

# 3 Primer to Schwinger-Keldysh EFT for open system

In the previous section, we have analyzed the low-energy spectrum of the domain-wall fluctuation in the dissipative Josephson junction. Starting from the stochastic sine-Gordon model, we have found that there is a notion of the symmetry and the corresponding conserved charge even though the physical charge is not conserved due to the dissipation. The derived effective Lagrangian (25) describes the dynamics of the fluctuation $\pi_{R/A}$ on the top of the domain-wall configuration. Considering two simple regimes (thin-wall and thick-wall regimes), we have shown the appearance of the NG mode in open systems: a pair of the diffusive gapless mode and gapped partner.

In the remaining part of the present paper, we investigate the universality of the obtained results, i.e., the consequences following just from the translational symmetry breaking in open systems, which is independent of the details of the microscopic model. To perform a model-independent analysis, we rely on the symmetry-based construction of a general effective Lagrangian based on the Schwinger-Keldysh formalism. This section is devoted to the preparation for writing down the general low-energy effective action in open systems.

## 3.1 Towards low-energy effective action for open quantum system

Let us begin with a brief review of the basics of the Schwinger-Keldysh effective field theory for open quantum systems (see, e. g., Ref. [76] for details). Suppose that the open system under consideration is realized as a subsystem of the closed total system. The total system then contains two kinds of dynamical degrees of freedom: system variables $\psi$ and environment variables $\sigma$ (see the left panel of Fig. 4). The goal of the Schwinger-Keldysh EFT for open systems is to describe $n$-point real-time correlation functions of low-energy observables $\hat{\mathcal{O}}(t, \boldsymbol{x})$ composed of the system variable $\psi$. The simplest example is an expectation value of the physical quantity $\hat{\mathcal{O}}(t, \boldsymbol{x})$ given by

$$\langle \hat{\mathcal{O}}(t, \boldsymbol{x}) \rangle \equiv \mathrm{Tr}\left[\hat{\rho}_0 \hat{\mathcal{O}}(t, \boldsymbol{x})\right] = \mathrm{Tr}\left[\hat{\rho}_0 \hat{U}^\dagger(t, -\infty) \hat{\mathcal{O}}(-\infty, \boldsymbol{x}) \hat{U}(t, -\infty)\right], \tag{48}$$

where $\hat{\rho}_0$ denotes an initial density operator at $t = -\infty$ and $\hat{U}(t, -\infty)$ does the time evolution operator of the total system from time $t = -\infty$ to time $t$. Note that the time evolution is generated by the unitary operator $\hat{U}(t, -\infty)$ since the total quantum system is assumed to be closed.

To systematically compute general $n$-point correlation functions for the system variable $\psi$, it is useful to introduce the closed-time-path generating functional. The closed-time-path gen-

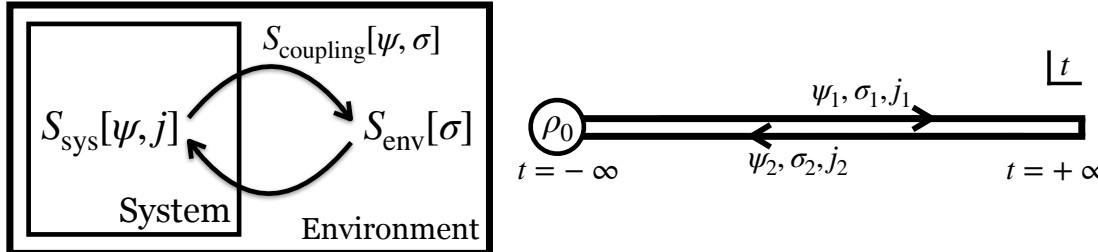

Figure 4: (Left) An open system realized through the interaction with the environment. (Right) Closed time contour in the Schwinger-Keldysh formalism.

erating functional is defined by putting the system in the presence of the different background fields $j_a$ ($a = 1, 2$) for the forward and backward time evolutions as (see the right panel of Fig. 4)

$$
\begin{aligned}
Z[j_1, j_2] &\equiv \mathrm{Tr}\left[\hat{\rho}_0 \hat{U}_{j_2}^{\dagger}(\infty, -\infty) \hat{U}_{j_1}(\infty, -\infty)\right] \\
&= \int \mathcal{D}\psi_1 \mathcal{D}\psi_2 \mathcal{D}\sigma_1 \mathcal{D}\sigma_2 \, \exp\left(\mathrm{i}S_{\mathrm{tot}}[\psi_1, \sigma_1; j_1] - \mathrm{i}S_{\mathrm{tot}}[\psi_2, \sigma_2; j_2]\right)\rho_0(\psi, \sigma),
\end{aligned}
\tag{49}
$$

where $\hat{U}_j(\infty, -\infty)$ denotes the time-evolution operator with the external field $j(t, \boldsymbol{x})$, and $\rho_0(\psi, \sigma)$ is the initial probability weight determined by $\hat{\rho}_0$. Here, $S_{\mathrm{tot}}[\psi, \sigma; j]$ is a total microscopic action, which can be decomposed into the following three pieces

$$
S_{\mathrm{tot}}[\psi, \sigma; j] = S_{\mathrm{sys}}[\psi; j] + S_{\mathrm{env}}[\sigma] + S_{\mathrm{coupling}}[\psi, \sigma],
\tag{50}
$$

where $S_{\mathrm{sys}}[\psi; j]$ and $S_{\mathrm{env}}[\sigma]$ are the actions for the system and environment sectors while $S_{\mathrm{coupling}}[\psi, \sigma]$ describes the coupling between them. We assume the external field $j$ to be coupled only with the system variable $\psi$. Due to the two time-evolution operators $\hat{U}_{j_1}$ and $\hat{U}_{j_2}^{\dagger}$, the number of fields for the path-integral expression in Eq. (49) is doubled as given by, e.g., $\psi_1$ and $\psi_2$.

By integrating out the environment variables $\sigma$, we obtain the following path-integral formula for the generating functional of the system:

$$
Z[j_1, j_2] = \int \mathcal{D}\psi_1 \mathcal{D}\psi_2 \, \exp\left(\mathrm{i}S_{\mathrm{open}}[\psi_1, \psi_2; j_1, j_2]\right),
\tag{51}
$$

where we defined the microscopic action for the open system $S_{\mathrm{open}}$ as a sum of the original system action $S_{\mathrm{sys}}[\psi; j]$ and the influence functional $\Gamma[\psi_1, \psi_2]$ [77,78]:

$$
S_{\mathrm{open}}[\psi_1, \psi_2; j_1, j_2] = S_{\mathrm{sys}}[\psi_1; j_1] - S_{\mathrm{sys}}[\psi_2; j_2] + \Gamma[\psi_1, \psi_2],
\tag{52}
$$

where the influence functional $\Gamma[\psi_1, \psi_2]$ is defined as

$$
\begin{aligned}
\mathrm{e}^{\mathrm{i}\Gamma[\psi_1, \psi_2]} \equiv \int \mathcal{D}\sigma_1 \mathcal{D}\sigma_2 \\
\times \exp\left(\mathrm{i}S_{\mathrm{env}}[\sigma_1] + \mathrm{i}S_{\mathrm{coupling}}[\psi_1, \sigma_1] - \mathrm{i}S_{\mathrm{env}}[\sigma_2] - \mathrm{i}S_{\mathrm{coupling}}[\psi_2, \sigma_2]\right)\rho_0(\psi, \sigma).
\end{aligned}
\tag{53}
$$

We assume that the environment is large enough so that its energy spectrum can be regarded as continuous. Then, the influence functional $\Gamma[\psi_1, \psi_2]$ generally have an imaginary part, which describes the dissipative dynamics of the system variable.[2]

---

[2]The large volume limit of the environment makes the recurrence time diverge so as to induce the dissipation of physical charges of the system into the environment [78].

To further focus on the low-energy dynamics of the system variables $\psi_a$, we introduce the low-energy Wilsonian effective action. The Wilsonian effective action is formally defined by identifying low-energy degrees of freedom $\pi$ such as NG fields and the high-energy gapped degrees of freedom $\Psi$, separating the field $\psi$ into $\psi = \{\pi, \Psi\}$, and integrating out $\Psi$ as follows:

$$\exp\Big(\mathrm{i}S_{\mathrm{eff}}[\pi_1, \pi_2; j_1, j_2]\Big) \equiv \int \mathcal{D}\Psi_1 \mathcal{D}\Psi_2 \, \exp\Big(\mathrm{i}S_{\mathrm{open}}[\pi_1, \Psi_1, \pi_2, \Psi_2; j_1, j_2]\Big), \qquad (54)$$

where we introduced the doubled NG field (gapped field) as $\pi_1$ and $\pi_2$ ($\Psi_1$ and $\Psi_2$). The NG fields $\pi$ often correspond to a collective excitation, whose explicit expression in terms of the microscopic variables $\psi$ could be complicated. Thus, the first-principle derivation of $S_{\mathrm{eff}}[\pi_1, \pi_2; j_1, j_2]$ sketched above is difficult to accomplish in practice.

Despite the difficulty of the direct derivation, we have a practically sufficient symmetry-based approach to construct the Wilsonian effective action. The crucial point here is that the effective action $S_{\mathrm{eff}}[\pi_1, \pi_2; j_1, j_2]$ has to respect the symmetry of the action $S_{\mathrm{open}}[\psi_1, \psi_2; j_1, j_2]$, which allows us to formulate the systematic construction of $S_{\mathrm{eff}}[\pi_1, \pi_2; j_1, j_2]$. Thus, we need to pay attention to the symmetry structure of open systems, which is a little complicated due to the field doubling and the presence of the influence functional [49–53]. Moreover, the effective action $S_{\mathrm{eff}}[\pi_1, \pi_2; j_1, j_2]$ must satisfy not only the symmetry constraints but also some basic conditions resulting from the structure of the closed-time-path generating functional (49). In the following, we will briefly summarize these conditions (see, e.g., Refs. [40,41] for a detailed discussion in the case of the closed system).

## 3.2 Symmetry structure in open system

Suppose that the total microscopic action $S_{\mathrm{tot}}[\psi_a, \sigma_a; j_a = 0]$, including the environment variable, enjoys a continuous global $G$-symmetry, which acts on the system and environment fields as

$$\psi_a \to \psi_a + \epsilon_a \delta\psi_a, \quad \sigma_a \to \sigma_a + \epsilon_a \delta\sigma_a, \qquad (55)$$

where $\epsilon_a$ ($a = 1, 2$) denotes independent infinitesimal transformation parameters. For simplicity, we drop the external source from now on, whose inclusion is straightforward. Then, one finds $\mathrm{i}S_{\mathrm{tot}}[\psi_1, \sigma_1; j_1 = 0] - \mathrm{i}S_{\mathrm{tot}}[\psi_2, \sigma_2; j_2 = 0]$ is invariant under the doubled symmetry $G_1 \times G_2$, whose transformation parameters are given by $\epsilon_1$ and $\epsilon_2$, respectively.

From the above observation, one may expect that the action $S_{\mathrm{open}}[\psi_1, \psi_2]$ is also invariant under $(G_1 \times G_2)$-transformation. However, this is not true because there is a contribution coming from the influence functional $\Gamma[\psi_1, \psi_2]$. Due to the elimination of the environment variable, $\Gamma[\psi_1, \psi_2]$ generally induces mixing between the system fields with different subscripts $a = 1, 2$. As a result, the action $S_{\mathrm{open}}[\psi_1, \psi_2]$ is only invariant under the diagonal subgroup of $G_1 \times G_2$ generated by the transformation (55) with $\epsilon_1 = \epsilon_2(\equiv \epsilon_A)$ [49–53]:

$$S_{\mathrm{open}}[\psi_1, \psi_2] = S_{\mathrm{open}}[\psi_1 + \epsilon_A \delta\psi_1, \psi_2 + \epsilon_A \delta\psi_2]. \qquad (56)$$

We call this symmetry as the $G_A$-symmetry, or the $A$-type $G$-symmetry. In other words, the non-diagonal part of $G_1 \times G_2$ generated by $\epsilon_1 = -\epsilon_2$ is not the symmetry of the open system. This explicit symmetry breaking represents the violation of the conservation law for the physical charges: the physical charges of the open system are exposed to irreversible dissipation into the environment.

The crucial point here is that the open system still enjoys the $G_A$-symmetry despite the violating conservation law for the physical charges. As a result, a steady state of the open system can further break the remaining $G_A$-symmetry down to its subgroup $H_A$. Here, we

define the spontaneous $G_A$-symmetry breaking by the presence of the physical order parameter field $\Phi_R(t, \boldsymbol{x}) \equiv [\Phi_1(t, \boldsymbol{x}) + \Phi_2(t, \boldsymbol{x})]/2$ as follows:

$$\exists \langle \Phi_R(t, \boldsymbol{x}) \rangle \quad \text{such that} \quad \langle \delta \Phi_R(t, \boldsymbol{x}) \rangle \neq 0, \tag{57}$$

where $\delta \Phi_R$ represents the $G_A$-transformation of the order parameter, and $\langle \cdots \rangle$ denotes the path-integral average. We note that the order parameter field $\Phi_R(t, \boldsymbol{x})$ could be a composite operator of the system variable. In the classical stochastic limit of quantum open systems, this definition agrees with that introduced in Eq. (19) in the previous section.

In short, the possible symmetry structure in the open system is summarized as follows:

$$\begin{aligned} G_1 \times G_2 \;\; &\to \;\; G_A \qquad \text{(Explicit breaking by environment)} \\ &\to \;\; H_A \qquad \text{(Spontaneous breaking by stationary solution).} \end{aligned} \tag{58}$$

To grasp this symmetry structure, it may be useful to recall the result obtained in the previous section: the sine-Gordon model without the dissipation and noise has the doubled translational symmetry generated by the physical momentum $P_{i,R}$ and the auxiliary momentum $P_{i,A}$. The presence of the dissipation and noise makes $P_{i,R}$ nonconserved quantity, so that it explicitly breaks the nondiagonal part of the doubled translational symmetry. Besides, the domain-wall configuration further breaks the remaining translational symmetry, or $P_{x,A}$-symmetry, generated by $P_{x,A}$.

### 3.3 Requirements to Schwinger-Keldysh effective action

Due to the symmetry structure of open systems, it is convenient to employ the Keldysh basis, in which the doubled field is expressed as the sum and difference of the original ones. For example, we introduce the doubled NG fields in the Keldysh basis as

$$\pi_R(t, \boldsymbol{x}) \equiv \frac{\pi_1(t, \boldsymbol{x}) + \pi_2(t, \boldsymbol{x})}{2}, \quad \pi_A(t, \boldsymbol{x}) \equiv \pi_1(t, \boldsymbol{x}) - \pi_2(t, \boldsymbol{x}), \tag{59}$$

where $\pi_1$ and $\pi_2$ are the NG fields embedded in the order parameter field in the original (or 12) basis. This basis is useful because it separates the full dynamics into its classical part (averages) described by $R$-type fields, and its quantum part (fluctuations) described by $A$-type fields. As we elaborate shortly, we will focus on the classical stochastic regime of the NG field, which is defined by the effective Lagrangian composed of the terms up to two $\pi_A$. It is worth emphasizing that this truncation does *not* mean the original model needs to be the classical stochastic systems (like the dissipative Josephson junction discussed in the previous section).

Resulting from the structure of the Schwinger-Keldysh formalism, there are additional requirements to the low-energy Schwinger-Keldysh effective action. Following the discussion developed in Refs. [40, 41] in the analysis on closed systems, we require the effective action $S_{\text{eff}}[\pi_R, \pi_A]$ to satisfy the following conditions (see Refs. [40, 41] for derivation in detail):

1. **Unitarity condition** The generating functional $Z[j_1, j_2]$ satisfies $Z[j_1 = j, j_2 = j] = 1$ with the initial density operator $\hat{\rho}_0$ satisfying $\text{Tr}\,\hat{\rho}_0 = 1$. To respect this property, the effective action is assumed to satisfy

$$S_{\text{eff}}[\pi_1 = \pi_2 = \pi] = S_{\text{eff}}[\pi_R, \pi_A = 0] = 0. \tag{60}$$

   Since $Z[j_1 = j, j_2 = j] = 1$ follows from the unitarity of the time-evolution operator $\hat{U}_j^\dagger \hat{U}_j = 1$ for the total system, we call this as the unitarity condition.

2. **Conjugate condition** Taking complex conjugate of the generating functional, one finds $Z[j_1, j_2]^* = Z[j_2, j_1]$. To respect this condition, we require the effective action to satisfy

$$(S_{\text{eff}}[\pi_1, \pi_2])^* = -S_{\text{eff}}[\pi_2, \pi_1] \quad \Longleftrightarrow \quad (S_{\text{eff}}[\pi_R, \pi_A])^* = -S_{\text{eff}}[\pi_R, -\pi_A]. \tag{61}$$

3. **Convergent condition** In order to have a well-defined (or convergent) $Z[j_1, j_2]$, the imaginary part of the effective action is assumed to satisfy the following condition:

$$\text{Im} \, S_{\text{eff}}[\pi_R, \pi_A] \geq 0. \tag{62}$$

Notice that these conditions are quite general; i.e., they follow from the unitarity of the time-evolution operator, the self-adjointness and normalization condition of the initial density operator $\hat{\rho}_0$, and the stability of the steady state. We, however, note that there could be a class of open systems violating some of these properties: for instance, the convergent condition could be violated if open systems under consideration have no stable steady state. Thus, it may be fair to say that we focus on a simple class of open systems satisfying the above requirement like the dissipative Josephson junction system discussed in Sec. 2.

## 4 General analysis based on Schwinger-Keldysh EFT

In this section, we consider a general open system whose steady state spontaneously breaks the translational symmetry along the $x$-direction as in the dissipative sine-Gordon model with noise discussed in Sec. 2. We construct the most general effective action based only on the symmetry-breaking patterns and the Schwinger-Keldysh constraints introduced in the previous section. After identifying the NG fields in Sec. 4.1, we summarize the symmetries of the system in Sec. 4.2. Then, using the power counting scheme specified in Sec. 4.3, we write down the Wilsonian Schwinger-Keldysh effective action of the translational NG fields in Sec. 4.4. Restricting two simple regimes (thin-wall and thick-wall regimes), we investigate the dispersion relations of the resulting NG fields in Sec. 4.5. We also discuss the peculiar coupling term that could induce the KPZ universality class in Sec. 4.6.

### 4.1 NG field and material coordinate field

Let us consider a steady state of $(d+1)$-dimensional open systems that spontaneously breaks the $A$-type spatial translational symmetry along the $x$-direction, which we call the $P_{x,A}$-symmetry. In that situation, regardless of detailed information on the underlying microscopic theory, we have an order parameter characterizing the broken $P_{x,A}$-symmetry. In other words, following a general condition of spontaneous $G_A$-symmetry breaking in Eq. (57), we consider a condensate of the scalar order operator $\Phi(t, \boldsymbol{x})$ with $x$-coordinate dependence:[3]

$$\langle \Phi_R(t, \boldsymbol{x}) \rangle = \bar{\Phi}(x) \quad \text{with} \quad \partial_x \bar{\Phi}(x) \neq 0. \tag{63}$$

This condition matches with the definition of the spontaneous symmetry breaking (19) discussed in the previous section.

Relying on the existence of the inhomogeneous condensate (63), we introduce the doubled NG fields $\pi_1(t, \boldsymbol{x})$ and $\pi_2(t, \boldsymbol{x})$ as embedded fluctuations on the top of the steady-state configuration:

$$\Phi_1(t, \boldsymbol{x}) = \bar{\Phi}(x + \pi_1(t, \boldsymbol{x})) \quad \text{and} \quad \Phi_2(t, \boldsymbol{x}) = \bar{\Phi}(x + \pi_2(t, \boldsymbol{x})). \tag{64}$$

This embedding motivates us to define the doubled material (or Lagrangian) coordinate fields $X_1(t, \boldsymbol{x})$ and $X_2(t, \boldsymbol{x})$ as

$$X_1(t, \boldsymbol{x}) = x + \pi_1(t, \boldsymbol{x}) \quad \text{and} \quad X_2(t, \boldsymbol{x}) = x + \pi_2(t, \boldsymbol{x}), \tag{65}$$

---

[3]Although it is interesting to consider the inhomogeneous spinful condensate as discussed in Ref. [21] for the zero-temperature case, the consideration of that is beyond the scope of the present paper.

which enables us to interpret the NG fields $\pi_1(t, \boldsymbol{x})$ and $\pi_2(t, \boldsymbol{x})$ as the doubled one-dimensional displacement vectors in elastic theory [79]. In the following analysis, it is useful to introduce these variables in the Keldysh basis as

$$X_R(t, \boldsymbol{x}) \equiv \frac{X_1(t, \boldsymbol{x}) + X_2(t, \boldsymbol{x})}{2} = x + \pi_R(t, \boldsymbol{x}), \tag{66a}$$

$$\pi_A(t, \boldsymbol{x}) = X_1(t, \boldsymbol{x}) - X_2(t, \boldsymbol{x}), \tag{66b}$$

with $\pi_R(t, \boldsymbol{x})$ and $\pi_A(t, \boldsymbol{x})$ defined in Eq. (59). While we will use the NG field $\pi_R$ to investigate the spectrum of the domain-wall fluctuation, the material coordinate field $X_R$ will be more useful to construct the effective action by virtue of their simple transformation rules. This is because the material coordinates behave as a scalar even under the act of broken $P_{x,A}$-transformation, as we will see shortly. We also note that $\pi_A(t, \boldsymbol{x})$ is expected to be accompanied by the derivative of the order parameter $\bar{\Phi}'(x + \pi_R(t, \boldsymbol{x}))$. This is indeed the case if all $\pi_A(t, \boldsymbol{x})$ appears by expanding Eq. (64) on the top of the averaged position as

$$\Phi_1(t, \boldsymbol{x}) - \Phi_2(t, \boldsymbol{x}) = \bar{\Phi}'(x + \pi_R(t, \boldsymbol{x}))\pi_A(t, \boldsymbol{x}) + O(\pi_A^3), \tag{67}$$

where we can neglect the higher-order $O(\pi_A^3)$ terms in the classical stochastic limit. Note that this corresponds to the second equation in Eq. (22) in the dissipative Josephson junction.

## 4.2 Symmetries of the system

We assume that the underlying open system action $S_{\text{open}}$ enjoys symmetry under the $A$-type spacetime translation and spatial rotation, but not necessarily the Lorentz boost nor Galilean boost. Then, the effective action $S_{\text{eff}}[\pi_R, \pi_A]$ is also invariant under the act of these symmetries, defined by the diagonal parts of those transformations given in Eq. (56). Since the condensate fields $\Phi_1$ and $\Phi_2$ are assumed to be scalar quantities, the $A$-type translation and rotation act on the material coordinate field $X_R$ and the NG field $\pi_A$ as

$$\begin{cases} X_R(t, \boldsymbol{x}) \to X_R'(t, \boldsymbol{x}) = X_R(t + \epsilon_A^0, \boldsymbol{x} + \boldsymbol{\epsilon}_A), \\ \pi_A(t, \boldsymbol{x}) \to \pi_A'(t, \boldsymbol{x}) = \pi_A(t + \epsilon_A^0, \boldsymbol{x} + \boldsymbol{\epsilon}_A), \end{cases} \tag{68}$$

and

$$\begin{cases} X_R(t, \boldsymbol{x}) \to X_R'(t, \boldsymbol{x}) = X_R(t, \mathcal{R}_A^{-1}\boldsymbol{x}), \\ \pi_A(t, \boldsymbol{x}) \to \pi_A'(t, \boldsymbol{x}) = \pi_A(t, \mathcal{R}_A^{-1}\boldsymbol{x}), \end{cases} \tag{69}$$

with $\mathcal{R}_A \in \text{SO}(d)_A$. It is worth emphasizing that the material coordinate $X_R$ and the $A$-type NG field $\pi_A$ behave as scalars under the transformations while the $R$-type NG field $\pi_R$ transforms nonlinearly under the spatial translation along the $x$-axis as

$$\pi_R(t, \boldsymbol{x}) \to \pi_R'(t, \boldsymbol{x}) = \pi_R(t, \boldsymbol{x} + \boldsymbol{\epsilon}_A) + \epsilon_A^1. \tag{70}$$

This nonlinear transformation rule is a manifestation that $\pi_R$ defines the NG field corresponding to the broken $P_{x,A}$-symmetry. Thus, it is convenient to construct the effective action in terms of $X_R$ and $\pi_A$ instead of $\pi_R$ and $\pi_A$ to be consistent with the symmetries.

## 4.3 Power counting scheme

While there appears an infinite number of terms allowed even under the constraints resulting from both the symmetry and the structure of the Schwinger-Keldysh formalism, they can be systematically organized using an appropriate power counting scheme. As usual for the Schwinger-Keldysh EFT, we here employ a double expansion scheme: one for a derivative expansion justified in the low-energy limit, and the other for a fluctuation expansion assuming

the smallness of the fluctuation (see, e.g., Refs. [40, 41] for the detailed discussion on the fluctuation expansion).

To implement the double expansion schemes, it is useful to introduce two bookkeeping small parameters $p$ and $\mathcal{A}$: $p$ denotes a typical momentum scale at the scale of interest, and thus, assumed to be small at a low-energy regime, and $\mathcal{A}$ does the magnitude of the fluctuation attached to the $A$-type fields. Using these parameters, we employ the power counting scheme defined by

$$\partial_\mu^n \pi_R(t, \boldsymbol{x}) = O(p^n, \mathcal{A}^0), \qquad \partial_\mu^n \pi_A(t, \boldsymbol{x}) = O(p^n, \mathcal{A}^1), \tag{71}$$

with $\mu = 0, 1, \cdots, d$ and $n = 0, 1, 2, \cdots$, where $\mu = 0$ denotes the temporal index. We regard the momentum scale of the domain-wall fluctuation $\pi_R$ as small. In the effective action, not only the NG fields but also the coordinate $x$ can explicitly appear through $X_R$. Note that we also count the coordinate $x$ and its derivative as

$$x = O(p^0, \mathcal{A}^0), \qquad \partial_\mu x = \delta_\mu^1 = O(p^0, \mathcal{A}^0). \tag{72}$$

It should be emphasized that we do not assign a specific order to the derivative itself because its power counting is defined together with the objects on which it acts. We can translate our power counting scheme (71) in terms of the material coordinate $X_R$ as

$$\partial_t^n X_R = O(p^n, \mathcal{A}^n), \quad \partial_i^n X_R = \begin{cases} O(p^0, \mathcal{A}^0) & (n = 0, 1), \\ O(p^n, \mathcal{A}^0) & (n \geq 2). \end{cases} \tag{73}$$

The power counting of the spatial derivative acting on $X_R$ is a little complicated because the vector $\partial_i X_R$ contains an $O(p^0, \mathcal{A}^0)$ term. Accordingly, to write down all possible terms in the effective action, we will use $(\boldsymbol{\nabla} X_R)^2 - 1 = O(p^1, \mathcal{A}^0)$ as a building block instead of $(\boldsymbol{\nabla} X_R)^2$, which allows us to keep $X_R$ as the only scalar in $O(p^0, \mathcal{A}^0)$.

## 4.4 Constructing the effective action

Based on the above preparation, we now construct the effective action of the translational NG fields in open systems within the double expansion for $p$ and $\mathcal{A}$. In this paper, we restrict ourselves to construct the effective action up to the second order with respect to both $p$ and $\mathcal{A}$. We refer to this regime as a classical stochastic regime because the resulting effective action corresponds to the MSR action describing the stochastic dynamics of the NG fields.

Since only $\pi_A$ increases the order $\mathcal{A}$ in our power counting scheme, we first expand the effective action for $\mathcal{A}$ as

$$S_{\text{eff}}[\pi_R, \pi_A] = \int \mathrm{d}t \mathrm{d}^d x \, \mathcal{L}_{\text{eff}}(t, \boldsymbol{x}), \tag{74}$$

with

$$\mathcal{L}_{\text{eff}}(t, \boldsymbol{x}) = \pi_A(t, \boldsymbol{x}) \mathcal{F}_1[X_R(t, \boldsymbol{x})] + \frac{\mathrm{i}}{2} \pi_A(t, \boldsymbol{x}) \mathcal{F}_2[X_R(t, \boldsymbol{x}), \partial_\mu] \pi_A(t, \boldsymbol{x}) + O(\mathcal{A}^3), \tag{75}$$

where we introduced a function $\mathcal{F}_1[X_R(t, \boldsymbol{x})]$ of $X_R$ and its derivatives and a linear operator $\mathcal{F}_2[X_R(t, \boldsymbol{x}), \partial_\mu]$ that includes the derivative acting on $\pi_A$ on their right side.[4] Here, note that the effective Lagrangian at least contains one $\pi_A$ to satisfy the unitarity condition (60), and the first-order term with respect to $\pi_A$ is real and the second-order term is pure imaginary due to the conjugate condition (61). The first-order term gives the deterministic part of the equation of motion, while the second-order term describes the intensity of the noise added to the deterministic contribution, as we have shown in the MSR action (25) in Sec. 2.

---

[4]Since we are considering the effective action, we can give the first-order term for $\pi_A$ without the derivative term acting on $\pi_A$ with the help of the integration by parts.

We shall write down $\mathcal{F}_1$ and $\mathcal{F}_2$ relying on the derivative expansion. Let us first consider all possible terms up to $O(p^2)$ included in the single $A$-type field sector, or $\mathcal{F}_1$-term, which we identify as follows:

$$
\begin{aligned}
O(p^0): \;& f(X_R), \\
O(p^1): \;& \gamma(X_R)\partial_t X_R, \quad \lambda_s(X_R)[(\boldsymbol{\nabla}X_R)^2 - 1], \\
O(p^2): \;& f_t(X_R)\partial_t^2 X_R, \quad f_s(X_R)\boldsymbol{\nabla}^2 X_R, \quad f_{tx}(X_R)\partial_t(\boldsymbol{\nabla}X_R)^2, \quad f_x(X_R)\partial_i X_R\partial_j X_R\partial_i\partial_j X_R, \\
& \lambda_t(X_R)(\partial_t X_R)^2, \quad \lambda_{tx}(X_R)(\partial_t X_R)[(\boldsymbol{\nabla}X_R)^2 - 1], \quad \lambda_x(X_R)[(\boldsymbol{\nabla}X_R)^2 - 1]^2,
\end{aligned}
\tag{76}
$$

where summations over repeated spatial indices ($i = 1, 2, \ldots, d$) are assumed. Here, $f, \gamma, f_\alpha$, and $\lambda_\alpha$ with $\alpha = t, s, x, tx$ are certain real functions of $X_R$. We can obtain $\mathcal{F}_1$ by summing up all the terms in Eq. (76).

Let us continue to write down the second-order terms in the expansion with $\mathcal{A}$. Using the same strategy as above, we find 14 independent terms up to $O(p^2)$. In particular, only five of them are identified to have a quadratic term for the NG fields $\pi_R$ and $\pi_A$ when we expand them with respect to $\pi_R$. The parent terms of them are given by

$$
\begin{aligned}
O(p^0): \;& A(X_R), \\
O(p^2): \;& \kappa_t(X_R)\partial_t^2, \quad \kappa_s(X_R)\boldsymbol{\nabla}^2, \quad \kappa_{tx}(X_R)\boldsymbol{\nabla}X_R\cdot\boldsymbol{\nabla}\partial_t, \quad \kappa_x(X_R)\partial_i X_R\partial_j X_R\partial_i\partial_j,
\end{aligned}
\tag{77}
$$

where $A$ and $\kappa_\alpha$ ($\alpha = t, s, x, tx$) denote certain real functions of $X_R$ that satisfy the convergent condition. The other 9 terms have the same form as $O(p^1)$ and $O(p^2)$ terms in Eq. (76) and are given by replacing each coefficient function accordingly.

Although we have written down all the terms up to $O(p^2, \mathcal{A}^2)$ consistent with the symmetries, the resulting effective action is too general: We have not imposed the condition ensuring that the domain-wall solution satisfies $\pi_R = \pi_A = 0$, or equivalently $X_R = x$ and $\pi_A = 0$. In fact, the equation of motion reads

$$
0 = \frac{\delta S_{\text{eff}}}{\delta \pi_A(t, \boldsymbol{x})} = \mathcal{F}_1[X_R(t, \boldsymbol{x}), \partial_\mu] + O(\pi_A),
\tag{78}
$$

which does not, in general, leads to the solution $X_R = x$ and $\pi_A = 0$ because of the term $f(X_R)$. We then impose $f(X_R) = 0$, which is equivalent to eliminate the tadpole term appearing in the effective Lagrangian.

By expressing the material coordinates by the NG fields $\pi_R$ and $\pi_A$, we eventually obtain the most general effective Lagrangian up to $O(p^2, \mathcal{A}^2)$ as

$$
\begin{aligned}
\mathcal{L}_{\text{eff}} = \Big[ & -\gamma(x + \pi_R)\partial_t\pi_R + \lambda_s(x + \pi_R)[2\partial_x\pi_R + (\boldsymbol{\nabla}\pi_R)^2] - f_t(x + \pi_R)\partial_t^2\pi_R \\
& + f_s(x + \pi_R)\boldsymbol{\nabla}^2\pi_R + 2f_{tx}(x + \pi_R)\partial_t\partial_x\pi_R + f_x(x + \pi_R)\partial_x^2\pi_R \\
& + \lambda_t(x + \pi_R)(\partial_t\pi_R)^2 + 2\lambda_{tx}(x + \pi_R)\partial_t\pi_R\partial_x\pi_R + 4\lambda_x(x + \pi_R)(\partial_x\pi_R)^2 \Big]\pi_A \\
& + \frac{\mathrm{i}}{2}\pi_A\Big[ A(x + \pi_R) + \kappa_t(x + \pi_R)\partial_t^2 + \kappa_s(x + \pi_R)\boldsymbol{\nabla}^2 + \kappa_{tx}(x + \pi_R)\partial_t\partial_x \\
& + \kappa_x(x + \pi_R)\partial_x^2 + \widetilde{\mathcal{F}}_2 \Big]\pi_A + O(p^3, \mathcal{A}^3),
\end{aligned}
\tag{79}
$$

where $\widetilde{\mathcal{F}}_2$ is the sum of the terms in $\mathcal{F}_2$ other than those in Eq. (77). For later convenience, we added a negative sign for $\gamma(x + \pi_R)\partial_t\pi_R$ and $f_t(x + \pi_R)\partial_t^2\pi_R$. We note that the functional forms of the coefficient functions $\ell_m \equiv \{f_\alpha, \gamma, \lambda_\alpha, A, \kappa_\alpha\}$ cannot be specified within the EFT approach. These functions serve as low-energy coefficients (or functions) of the effective theory. Their functional forms are, in principle, determined from microscopic information of

the system, in particular, the configuration of the inhomogeneous condensate $\bar{\Phi}(x)$. Nevertheless, the coefficient functions are not completely arbitrary within the EFT approach, and as we discuss later, their sign would be somewhat constrained by the stability of the steady state. It is worth emphasizing that the effective Lagrangian (79) captures the general nonlinear interaction between the fluctuation (or NG fields) at the low-energy $O(p^2)$ classical stochastic regime.

## 4.5 Dynamics of the NG mode

The resulting effective Lagrangian captures the low-energy fluctuation around the inhomogeneous condensate $\bar{\Phi}(x)$. To investigate the low-energy spectrum of the NG field, we pick up the quadratic-order part for $\pi_R$ and $\pi_A$ by expanding the effective Lagrangian (79) with respect to the NG fields as

$$\mathcal{L}_{\text{eff}} = \mathcal{L}_{\text{eff}}^{(2)} + \mathcal{L}_{\text{eff}}^{(\text{int})} + O(p^3, \mathcal{A}^3), \tag{80}$$

where we find the quadratic part of the effective Lagrangian as

$$\mathcal{L}_{\text{eff}}^{(2)} = \Big[ -\gamma(x)\partial_t \pi_R + 2\lambda_s(x)\partial_x \pi_R - f_t(x)\partial_t^2 \pi_R + f_s(x)\boldsymbol{\nabla}^2 \pi_R + 2f_{tx}(x)\partial_t \partial_x \pi_R$$

$$+ f_x(x)\partial_x^2 \pi_R \Big]\pi_A + \frac{\mathrm{i}}{2}\pi_A \Big[ A(x) + \kappa_t(x)\partial_t^2 + \kappa_s(x)\boldsymbol{\nabla}^2 + \kappa_{tx}(x)\partial_t \partial_x + \kappa_x(x)\partial_x^2 \Big]\pi_A. \tag{81}$$

By comparing the present result with Eq. (25), one identifies the low-energy coefficient functions in the dissipative Josephson junction as

$$\gamma(x) = \alpha\bar{\phi}'(x)^2 - \beta\bar{\phi}'(x)\bar{\phi}'''(x), \quad f_t(x) = f_s(x) = \bar{\phi}'(x)^2, \quad \lambda_s(x) = \bar{\phi}'(x)\bar{\phi}''(x),$$
$$f_{tx}(x) = \beta\bar{\phi}'(x)\bar{\phi}''(x), \quad A(x) = A\bar{\phi}'(x)^2, \tag{82}$$

where the others not shown in this equation are identified as zero.[5] Thus, the low-energy coefficients in the dissipative sine-Gordon kink are completely controlled by the functional form of the domain-wall solution $\bar{\phi}(x)$. As in the model analysis in Sec. 2, the $x$-dependence of the low-energy coefficients makes further analysis difficult, and we will consider two simplified situations in the following analysis: the thin-wall regime and the thick-wall regime.

### 4.5.1 Low-energy spectrum in thin-wall regime

As in the previous analysis given in Sec. 2, we rely on the ansatz that the NG fields are localized at the domain-wall position $x = 0$, and introduce the localized NG fields $\tilde{\pi}_R$ and $\tilde{\pi}_A$ as

$$\tilde{\pi}_{R/A}(t, \boldsymbol{x}_\perp) \equiv \pi_{R/A}(t, x = 0, \boldsymbol{x}_\perp) \quad \text{with} \quad \boldsymbol{x}_\perp = (x^2, x^3, \cdots, x^d). \tag{83}$$

Here, we define the averaged low-energy coefficients $\bar{\ell}_m = \{\bar{f}_\alpha, \bar{\gamma}, \bar{\lambda}_\alpha, \bar{A}, \bar{\kappa}_\alpha\}$ as

$$\bar{\ell}_m = \int_{-\infty}^{\infty} \mathrm{d}x\, \ell_m(x). \tag{84}$$

It should be noted that some of $\bar{\ell}_m$ could vanish through the averaging procedure: For instance, recall $\bar{\lambda}_s$ in the dissipative Josephson junction is zero though it is present before performing the integration. In the following analysis, we assume that all the coefficient $\bar{\ell}_m$ does not vanish to find the most general low-energy spectrum. We also assume the positivity of some low-energy

---

[5]The term $\beta\bar{\phi}'(x)^2\pi_A\partial_t\boldsymbol{\nabla}^2\pi_R$ in Eq. (25) is regarded as $O(p^3, \mathcal{A}^1)$ in our power counting so that there is no term in Eqs. (79) and (81) that can match it.

coefficients $\bar{f}_t$, $\bar{f}_s$, and $\bar{\gamma}$. We expect that this is the case for a large class of models since the first two coefficients are often proportional to the squared condensate while the last one represents the dissipative constant. However, we cannot show this assumption model-independently, and thus, the positivity of these coefficients should be regarded as our another assumption.

Substituting the above ansatz into Eq. (81), we obtain the effective Lagrangian at the quadratic order as

$$
\begin{aligned}
\mathcal{L}^{(2)}_{\text{thin}} &= \widetilde{\pi}_A(t,\boldsymbol{x}_\perp)\big[-\bar{f}_t\partial_t^2 - \bar{\gamma}\partial_t + \bar{f}_s\boldsymbol{\nabla}_\perp^2\big]\widetilde{\pi}_R(t,\boldsymbol{x}_\perp) + \frac{\mathrm{i}}{2}\widetilde{\pi}_A(t,\boldsymbol{x}_\perp)\big[\bar{A} + \bar{\kappa}_t\partial_t^2 + \bar{\kappa}_s\boldsymbol{\nabla}_\perp^2\big]\widetilde{\pi}_A(t,\boldsymbol{x}_\perp) \\
&= -\frac{1}{2}\big(\widetilde{\pi}_R(t,\boldsymbol{x}_\perp) \quad \widetilde{\pi}_A(t,\boldsymbol{x}_\perp)\big)\begin{pmatrix} 0 & G_{A;\perp}^{-1} \\ G_{R;\perp}^{-1} & G_{K;\perp}^{-1} \end{pmatrix}\begin{pmatrix} \widetilde{\pi}_R(t,\boldsymbol{x}_\perp) \\ \widetilde{\pi}_A(t,\boldsymbol{x}_\perp) \end{pmatrix},
\end{aligned}
\tag{85}
$$

where we defined $\boldsymbol{\nabla}_\perp \equiv \partial/\partial\boldsymbol{x}_\perp$.[6] Here, we also introduced the inverse of the Green's functions for the localized NG fields $\widetilde{\pi}_R$ and $\widetilde{\pi}_A$ as

$$
G_{R;\perp}^{-1}(t,\boldsymbol{x}_\perp) = \bar{f}_t\partial_t^2 + \bar{\gamma}\partial_t - \bar{f}_s\boldsymbol{\nabla}_\perp^2, \tag{86a}
$$

$$
G_{A;\perp}^{-1}(t,\boldsymbol{x}_\perp) = \bar{f}_t\partial_t^2 - \bar{\gamma}\partial_t - \bar{f}_s\boldsymbol{\nabla}_\perp^2, \tag{86b}
$$

$$
G_{K;\perp}^{-1}(t,\boldsymbol{x}_\perp) = -\mathrm{i}\big[\bar{A} + \bar{\kappa}_t\partial_t^2 + \bar{\kappa}_s\boldsymbol{\nabla}_\perp^2\big]. \tag{86c}
$$

The retarded Green's function (86a) allows us to extract the low-energy spectrum of the localized NG modes by solving

$$
0 = G_{R;\perp}^{-1}(\omega,\boldsymbol{k}_\perp) = -\bar{f}_t\omega^2 - \mathrm{i}\bar{\gamma}\omega + \bar{f}_s\boldsymbol{k}_\perp^2 \quad \text{with} \quad \boldsymbol{k}_\perp = (k^2, k^3, \cdots, k^d). \tag{87}
$$

As a result, we find the dispersion relation of the localized NG modes as

$$
\omega(\boldsymbol{k}_\perp) = \frac{-\mathrm{i}\bar{\gamma} \pm \mathrm{i}\sqrt{\bar{\gamma}^2 - 4\bar{f}_t\bar{f}_s\boldsymbol{k}_\perp^2}}{2\bar{f}_t} = \begin{cases} -\mathrm{i}\dfrac{\bar{f}_s}{\bar{\gamma}}\boldsymbol{k}_\perp^2 + O(\boldsymbol{k}_\perp^4), \\ -\mathrm{i}\dfrac{\bar{\gamma}}{\bar{f}_t} + \mathrm{i}\dfrac{\bar{f}_s}{\bar{\gamma}}\boldsymbol{k}_\perp^2 + O(\boldsymbol{k}_\perp^4). \end{cases} \tag{88}
$$

The dispersion relation derived here shows the essentially same behavior as that derived in Sec. 2 [recall Eq. (30)]. Likewise, the low-frequency and low-wavenumber part of the symmetric Green's function also shows the same behavior as demonstrated in Fig. 2. Here, we used our assumption, or the positivity of some low-energy coefficients $\bar{f}_t$, $\bar{f}_s$, and $\bar{\gamma}$. Thus, we conclude that the appearance of the paired mode—one gapless diffusion and one gapped diffusion—is universal in the thin-wall regime of the realized domain wall in open systems.

### 4.5.2 Low-energy spectrum in thick-wall regime

In the thick-wall regime, the fluctuation (NG field) cannot see that the slope (or derivative) of the condensate is changing. For this reason, we simply replace all coefficient functions in the effective Lagrangian (81) with constants. From the symmetry viewpoint, this replacement is understood as a consequence of the invariance under $X_R \to X_R - \epsilon$, which results from the emergent uniformity of the steady state in the thick-wall regime. This invariance is a symmetry about the reassignment of the material coordinate $X_R$, rather than a spatial translational symmetry, which prohibits the appearance of $X_R$ without derivatives in the effective Lagrangian.

---

[6]The action in the thin wall regime describes the motion of the membrane-like object. In closed systems, it can be represented by the Nambu-Goto action with an induced metric on the membrane (see, e.g., Ref. [21]). However, the domain-wall effective action in open systems does not allow such an expression because we cannot express the dissipative term in terms of the induced metric.

The resulting form of the retarded Green's function is found as

$$G_R^{-1}(\omega, \boldsymbol{k}) = -f_t \omega^2 - i\gamma\omega + f_s \boldsymbol{k}^2 + f_x k_x^2 - 2i\lambda_s k_x - 2f_{tx}\omega k_x, \tag{89}$$

with $\boldsymbol{k} = (k_x, \boldsymbol{k}_\perp)$. Note that the isotropy is not fully recovered even in the thick wall regime because the slope takes a non-zero constant value, so that anisotropic terms can survive.[7] It should be mentioned, nonetheless, that these anisotropic terms would not frequently appear due to the discrete symmetry discussed in the last paragraph of Sec. 4.6. Noting the presence of the anisotropy in the momentum space, we introduce the polar angle $\theta$ measured from the $k_x$-direction, which expresses the momentum along the $x$-direction as $k_x = |\boldsymbol{k}|\cos\theta$. Solving $G_R^{-1}(\omega, \boldsymbol{k}) = 0$, we find the anisotropic dispersion relation of the NG mode given by

$$\omega(\boldsymbol{k}) = \begin{cases} -\dfrac{2\lambda_s}{\gamma}|\boldsymbol{k}|\cos\theta - i\dfrac{f_s\gamma^2 + (f_x\gamma^2 + 4f_{tx}\lambda_s\gamma - 4f_t\lambda_s^2)\cos^2\theta}{\gamma^3}|\boldsymbol{k}|^2 + O(|\boldsymbol{k}|^3), \\ -i\dfrac{\gamma}{f_t} + 2\left(\dfrac{\lambda_s}{\gamma} - \dfrac{f_{tx}}{f_t}\right)|\boldsymbol{k}|\cos\theta + O(|\boldsymbol{k}|^2). \end{cases} \tag{90}$$

In contrast to the thin-wall regime, the dispersion relation (90) with nonvanishing $\lambda_s$ supports the propagating gapless mode with the momentum $k_x(= |\boldsymbol{k}|\cos\theta)$, along which the translational symmetry is broken. This propagating mode does not appear in the model analysis in Sec. 2 because the coefficient $\lambda_s$ vanishes in the dissipative sine-Gordon model in the thick-wall regime (see also the discussion in the subsequent section). Furthermore, it is also remarkable that this mode could cause instability even with $\gamma > 0$ since the dispersion relation can have a positive imaginary part at the soft momentum region. Since the maximum imaginary part appears when the momentum is along $x$-direction ($\theta = 0$), we see that the instability along, at least, $x$-direction takes place when the following condition is satisfied:

$$(f_s + f_x)\gamma^2 + 4f_{tx}\lambda_s\gamma - 4f_t\lambda_s^2 \leq 0, \tag{91}$$

where we used the assumption on the positive damping coefficient $\gamma > 0$.

Figure 5 shows the dispersion relation (90) for three different values of $\lambda_s$—two for the stable regimes and the other for the unstable regime—at three different polar angles $\theta = 0, \pi/4$, and $\pi/2$ measured from the direction along which the translation symmetry is broken. One clearly sees that the dispersion relation is anisotropic, and the lowest panel shows a possible appearance of the unstable mode along $x$-direction while the perpendicular direction does not support that. The anisotropic and potentially unstable behavior is remarkable in the sense that it does not appear in the case of the internal/time-translational symmetry breaking, nor the translational symmetry breaking in closed systems [49, 53] (see also Appendix A for the discussion of the domain-wall EFT in finite-temperature closed systems).

## 4.6 Kardar-Parisi-Zhang coupling constant $\bar{\lambda}_s$

The peculiar behavior of the thick-wall dispersion relation is caused by the coefficient $\lambda_s$. While it vanishes at the quadratic part of the effective Lagrangian in the thin-wall, we lastly remark that this coefficient $\lambda_s$ may play an important role even in the thin-wall regime via the interaction term.

In the thin-wall regime, the quadratic part of the effective Lagrangian given by Eq. (85) is equivalent to the following stochastic equation of motion:

$$[-\bar{f}_t \partial_t^2 - \bar{\gamma}\partial_t + \bar{f}_s \boldsymbol{\nabla}_\perp^2]\tilde{\pi}(t, \boldsymbol{x}_\perp) = \xi(t, \boldsymbol{x}_\perp), \tag{92}$$

---

[7]This is the same as the thick-wall regime of Sec. 2.4.2, where the wall thinness $m$ is not taken to zero.

(a) Stable regime with $(f_t, f_s, f_x, f_{tx}, \gamma, \lambda_s) = (1.0, 1.0, 0.5, 0.3, 1.0, 0.1)$

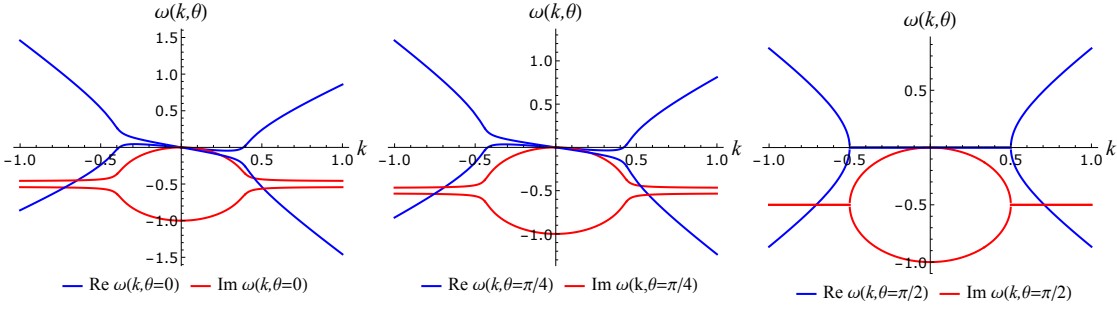

(b) Stable regime with $(f_t, f_s, f_x, f_{tx}, \gamma, \lambda_s) = (1.0, 1.0, 0.5, 0.3, 1.0, 0.5)$

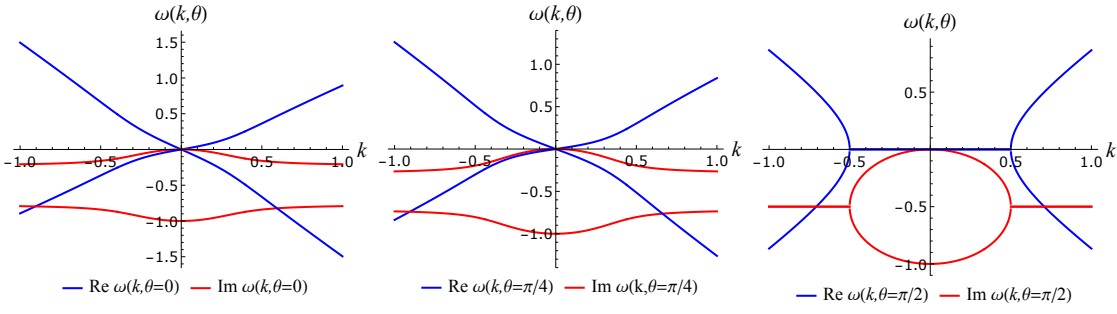

(c) Unstable regime with $(f_t, f_s, f_x, f_{tx}, \gamma, \lambda_s) = (1.0, 1.0, 0.5, 0.3, 1.0, 1.0)$

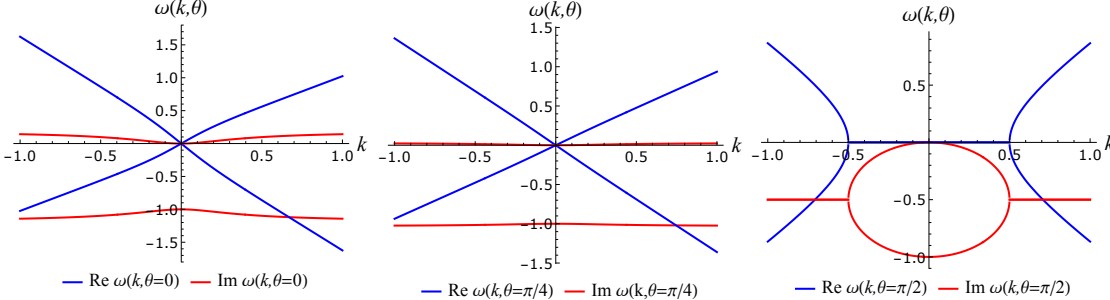

Figure 5: The dispersion relation at three different polar angles in (a)-(b) stable regimes (upper panels) and (c) an unstable regime (a lower panel).

with a noise $\xi(t, \boldsymbol{x}_\perp)$ obeying

$$\langle \xi(t, \boldsymbol{x}_\perp) \rangle_\xi = 0, \qquad \langle \xi(t, \boldsymbol{x}_\perp) \xi(t', \boldsymbol{x}'_\perp) \rangle_\xi = [\bar{A} + \bar{\kappa}_t \partial_t^2 + \bar{\kappa}_s \boldsymbol{\nabla}_\perp^2] \delta(t - t') \delta^{(d-1)}(\boldsymbol{x}_\perp - \boldsymbol{x}'_\perp). \tag{93}$$

By further focusing on the long-time and long-distance limit, we keep only the leading-derivative part, or set $\bar{f}_t$, $\bar{\kappa}_t$, and $\bar{\kappa}_s$ to zero. As a result, the above equation reduces to a linearized stochastic differential equation called the Edwards-Wilkinson equation, which describes the linear surface growth [24].

Let us then investigate the effects of nonlinear fluctuation. To incorporate this, we first expand Eq. (79) and keep all the cubic interaction terms in the original effective Lagrangian, which results in

$$\mathcal{L}_{\text{eff}}^{\text{int},(3)} = \pi_A \Big[ \pi_R \Big( -\gamma'(x) \partial_t + 2\lambda'_s(x) \partial_x - f'_t(x) \partial_t^2 + f'_s(x) \boldsymbol{\nabla}^2 + 2f'_{tx}(x) \partial_t \partial_x + f'_x(x) \partial_x^2 \Big) \pi_R$$
$$+ \lambda_s(x)(\boldsymbol{\nabla} \pi_R)^2 + \lambda_t(x)(\partial_t \pi_R)^2 + 2\lambda_{tx}(x) \partial_t \pi_R \partial_x \pi_R + 4\lambda_x(x)(\partial_x \pi_R)^2 \Big] + O(p^3, \mathcal{A}^2). \tag{94}$$

Here, we wrote down only the $\mathcal{F}_1$-term to illustrate the leading-order effect of the nonlinear fluctuation. We further simplify these terms by putting an assumption on the low-energy coefficients as $\ell_m(x = +\infty) = \ell_m(x = -\infty)$. This assumption enables us to drastically reduce the number of cubic interaction terms in the thin-wall regime thanks to

$$\int \mathrm{d}x \, \ell'_m(x) = \ell_m(x = +\infty) - \ell_m(x = -\infty) = 0. \tag{95}$$

Thus, the terms proportional to $\ell'_m(x)$ in Eq. (94) vanish, so that we only have the two cubic nonlinear interaction terms in the thin-wall regime as

$$\mathcal{L}^{\mathrm{int},(3)}_{\mathrm{thin}} = \widetilde{\pi}_A \big[ \bar{\lambda}_s (\boldsymbol{\nabla}_\perp \tilde{\pi}_R)^2 + \bar{\lambda}_t (\partial_t \tilde{\pi}_R)^2 \big] + O(p^3, \mathcal{A}^2). \tag{96}$$

Let us then focus on the long-time and long-distance limit again, and briefly discuss the possible universality class of the derived effective Lagrangian of Eqs. (85) and (96). First of all, we find $-\tilde{\pi}_A \bar{\gamma} \partial_t \tilde{\pi}_R$ as the leading temporal derivative term by assuming that $\bar{\gamma}$ does not vanish. Owing to this term, we can drop the $O(\partial_t^2)$-terms in the effective Lagrangian for the long-time dynamics, which means that we miss the gapped partner of the gapless diffusion mode. Besides, the nonvanishing $\bar{A}$ allows us to further drop all derivative terms controlling the magnitude of the frequency and wavenumber dependence of noise. We also note that we keep the leading-order terms in our double expansion scheme with respect to $p$ and $\mathcal{A}$. As a consequence, we find the following reduced effective Lagrangian:

$$\mathcal{L}_{\mathrm{thin}} = \widetilde{\pi}_A \big[ -\bar{\gamma} \partial_t + \bar{f}_s \boldsymbol{\nabla}_\perp^2 \big] \widetilde{\pi}_R + \bar{\lambda}_s \widetilde{\pi}_A (\boldsymbol{\nabla}_\perp \widetilde{\pi}_R)^2 + \frac{\mathrm{i}}{2} \bar{A} \widetilde{\pi}_A^2. \tag{97}$$

This effective Lagrangian precisely matches with the MSR effective Lagrangian for the KPZ equation defined by the nonlinear stochastic partial differential equation

$$-\bar{\gamma} \partial_t \widetilde{\pi}(t, \boldsymbol{x}_\perp) + \bar{f}_s \boldsymbol{\nabla}_\perp^2 \widetilde{\pi}(t, \boldsymbol{x}_\perp) + \bar{\lambda}_s \big( \boldsymbol{\nabla}_\perp \widetilde{\pi}(t, \boldsymbol{x}_\perp) \big)^2 = \xi(t, \boldsymbol{x}_\perp), \tag{98}$$

where $\xi(t, \boldsymbol{x}_\perp)$ denotes the Gaussian white noise satisfying

$$\langle \xi(t, \boldsymbol{x}_\perp) \rangle = 0, \quad \langle \xi(t, \boldsymbol{x}_\perp) \xi(t', \boldsymbol{x}'_\perp) \rangle = \bar{A} \delta(t - t') \delta^{(d-1)}(\boldsymbol{x}_\perp - \boldsymbol{x}'_\perp). \tag{99}$$

We thus specify that the term proportional to $\bar{\lambda}_s$ corresponds to the nonlinear term in the KPZ equation. Based on this result, we speculate that the original effective theory defined by Eqs. (85) and (96) belongs to the same universality class as those described by the KPZ equation [25]. In other words, the constructed effective theory is capable of capturing both the linear surface growth of the Edwards-Wilkinson equation [24] and the possible emergence of the KPZ universality class induced by the term proportional to $\bar{\lambda}_s$.[8]

In summary, the symmetry-based effective theory provides a derivation of the universal low-energy dynamics of the fluctuating domain wall, which is equivalent to the stochastic surface growth equation. The result of this section implies that the universality class of the domain-wall dynamics could be controlled by the presence of $\bar{\lambda}_s$ since it gives the KPZ nonlinear coupling. This is a remarkable property of open systems with spontaneous symmetry breaking since the cubic interaction proportional to the KPZ coupling $\bar{\lambda}_s$ is absent in the effective theory of the NG mode in closed systems (see Appendix A). However, it should be also emphasized that the appearance of the KPZ coupling is not guaranteed. For example, the KPZ coupling vanishes if the underlying dynamics is invariant under the discrete transformation

---

[8]Investigating the universality class with the help of the dynamic renormalization group approach is an interesting issue, but beyond the scope of this paper.

that exchanges the two different steady states separated by the domain wall. This exchanging transformation is typically realized as a sign inversion of the condensation field, which leads to the transformations $X_R \to -X_R$ and $\pi_A \to -\pi_A$. The invariance of the action to these transformations restricts $\lambda_s(x)$ to be the odd function $\lambda_s(x) = -\lambda_s(-x)$, so that the averaged coupling $\bar{\lambda}_s$ is shown to be zero. In the Josephson junction system, the MSR action (13) is invariant under the discrete transformations, $\phi_R \to 2\pi - \phi_R$ and $\phi_A \to -\phi_A$. (Recall that $\phi_R$ corresponds to a phase, and $\phi_R = 0$ and $\phi_R = 2\pi$ are equivalent.) This explains why the effective theory investigated in Sec. 2 lacks the KPZ coupling in the thin-wall regime.

# 5 Summary and outlook

In this paper, we have investigated the low-energy dynamics of the fluctuating domain wall in nonequilibrium open systems with the symmetry-based EFT. In Sec. 2, we have discussed the dissipative Josephson junction in $(2+1)$-dimensions, and introduced the notion of the symmetries in open systems and the MSR formalism to exploit them. We have then derived the MSR action for the fluctuations around the sine-Gordon kink, which describes a pair of the diffusive gapless mode and its gapped partner. Based on the constructed effective Lagrangian, we have also discussed experimental observables in the JTL. Section 3 has been devoted to the introduction of the low-energy Wilsonian effective action in the Schwinger-Keldysh formalism as preparation for discussing the universal consequences resulting from the translational symmetry breaking in open systems. In Sec. 4, we have derived the most general effective Lagrangian for the NG mode and its partner associated with the one-dimensional translational symmetry breaking in open systems. The thin-wall regime of the constructed effective theory confirmed that the emergence of the diffusive NG mode is a model-independent general consequence of the translational symmetry breaking. Moreover, we have also found a remarkable property of the possible term proportional to $\bar{\lambda}_s$, which is absent in the two simplified regimes of the Josephson junction system. We have shown that the term is peculiar to open systems, which could generate the KPZ nonlinear coupling in the thin-wall regime or cause the instability in the thick-wall regime. As a result, the macroscopic dynamics of the thin domain wall were likely to be controlled by the presence/absence of the KPZ coupling $\bar{\lambda}_s$.

There are several prospects from the present paper. While we have focused on the domain-wall dynamics in the dissipative Josephson junction, the similar domain-wall dynamics driven by the electric current plays an important role in magnetic materials (see, e.g., Ref. [80]). In this case, a nontrivial coupled dynamics of the domain-wall fluctuation and spin wave is expected to take place as is the case for closed systems [22], which arises as an interplay of the one-dimensional translational symmetry and spin-rotational symmetry. The use of the Landau-Lifshitz-Gilbert equation [81, 82] allows us to investigate their coupled dynamics.

It is also interesting to generalize our formulation into higher-dimensional or periodic variants of the translational symmetry breaking. While we mainly restrict ourselves to the one-dimensional domain wall, we can apply our formulation to the higher-dimensional system as well as the periodic configuration. While the effective theory of them—e.g., two-directional translational symmetry breaking by a vortex string [83–87] and skyrmion crystal [88–92]— has been attracting much attention, a little is known for their open system counterparts. Combining with the recent development of experimental techniques in, e.g., ultracold-atomic and magnetic systems, we can investigate their possible universal nonequilibrium dynamics in open systems. For that purpose, it is important to theoretically classify the dynamic universality class of the NG modes in open systems by taking into account the possible interaction term like KPZ coupling with the help of the dynamical renormalization group method [93–95].

## Acknowledgements

The authors thank Toshifumi Noumi for the collaboration in the early stage of this work. The authors also thank Tomoya Hayata and Yoshimasa Hidaka for valuable discussions.

**Funding information** K. F. was supported by RIKEN Junior Research Associate Program, by JSPS KAKENHI (Grant No. 19J13698). K. F is also supported by the Deutsche Forschungsgemeinschaft (DFG, German Research Foundation), project-ID 273811115 (SFB1225 ISOQUANT). M. H. was supported by the US Department of Energy, Office of Science, Office of Nuclear Physics under Award No. DE-FG0201ER41195. This work was partially supported by the RIKEN iTHEMS Program, in particular iTHEMS Non-Equilibrium Working group.

## A  Effective Lagrangian for domain wall in closed systems

In this appendix, we construct the effective field theory of the domain wall in finite-temperature closed systems and present qualitative differences with the result in open systems. The complete analysis requires consideration of the hydrodynamic mode [38–44], but we here only focus on the domain-wall degrees of freedom. There are two main sources making the distinction between open systems and closed systems: the symmetry structure and additional Schwinger-Keldysh constraint corresponding to the Kubo-Martin-Schwinger (KMS) condition [96, 97]. After explaining these two new ingredients, we construct the leading-order general effective Lagrangian and investigate the energy spectrum.

**Symmetry structure of closed system.** First of all, we define closed systems as the systems in which the physical (or $R$-type) charges obey the conservation laws. In other words, we do not separate the system and environment so that the closed-time-path generating functional takes the form of the second line in Eq. (49) (we consider both $\psi$ and $\sigma$ as dynamical degrees of freedom). Then, one finds that $S_{\text{tot}}[\psi_1, \sigma_1] - S_{\text{tot}}[\psi_2, \sigma_2]$ enjoys two symmetries defined by Eq. (55) with independent parameters $\epsilon_1$ and $\epsilon_2$. Thus, it is tempting to say that the system enjoys the doubled symmetry $G_1 \times G_2$, but this is *not* true.

To see this, we turn our attention to the initial density operator $\rho_0(\psi, \sigma)$, which defines a boundary condition for $\psi_1, \sigma_1$ and $\psi_2, \sigma_2$. The crucial point here is that the nondiagonal part of $G_1 \times G_2$ defined by $\epsilon_1 = -\epsilon_2 = \epsilon_A/2$ breaks this boundary condition. Since the initial state breaks the symmetry while the action $S_{\text{tot}}[\psi_1, \sigma_1] - S_{\text{tot}}[\psi_2, \sigma_2]$ preserves it, we can interpret this as a variant of the spontaneous symmetry breaking. Thus, the nondiagonal symmetry in the Schwinger-Keldysh formalism is always spontaneously broken even if the system respects a conservation law for the physical charge.

In summary, the possible symmetry structure in the closed system is given by

$$
\begin{aligned}
G_1 \times G_2 \;\to\; & G_A && \text{(Spontaneous breaking by boundary condition)} \\
\to\; & H_A && \text{(Spontaneous breaking by stationary solution)},
\end{aligned}
\tag{A.1}
$$

instead of Eq. (58) in open systems. In contrast to open systems, we now regard the first part $G_1 \times G_2 \to G_A$ as the spontaneous symmetry breaking, and thus the low-energy effective theory needs to respect the nondiagonal part of $G_1 \times G_2$ as well as the diagonal one. This symmetry structure is what the effective field theory of a dissipative fluid respects (see, e.g., Refs. [40, 41]).

Suppose that the closed system under consideration realizes a stationary state, which breaks the diagonal part of one-dimensional spatial translational symmetry along the

$x$-direction. In other words, the system supports the inhomogeneous condensate (63), from which we define the doubled NG fields and material coordinate fields as embedding [recall the discussion around Eqs. (64)-(66)]. Now, we need to respect the spontaneously broken nondiagonal part of $G_1 \times G_2$. This is accomplished by requiring the shift symmetry for $\pi_A$ since $\pi_A$ transforms nonlinearly as $\pi_A \to \pi_A + \epsilon_A$ under that symmetry. As a result, the invariant building blocks used to construct the EFT of the closed-system domain wall are given by

$$X_R(t, \boldsymbol{x}), \quad \partial_t \pi_A(t, \boldsymbol{x}), \quad \partial_i \pi_A(t, \boldsymbol{x}) \quad \text{and their derivatives.} \tag{A.2}$$

Possible terms appearing in the closed system domain-wall EFT is clearly restricted compared with the open system one; $\pi_A$ needs to be accompanied by the derivative. As for the power-counting scheme, we employ the same one with that defined in the main text, which forces us to be careful since the spatial derivative of $X_R(t, \boldsymbol{x})$ contains the mixed-order contribution. We, however, focus only on the leading-order part to illustrate qualitative differences with the open system result in the main text.

**Dynamical KMS symmetry.** If we assume that the initial density operator is given by a thermal density operator, there is another Schwinger-Keldysh constraint for the closed system, called the KMS condition [96, 97]. The KMS condition is the identity, which holds for closed systems staying initially in the thermal state. Since the closed-time-path generating functional for such systems also satisfies a variant of the KMS condition, the low-energy effective theory needs to be a consistent theory reproducing the KMS condition.

To respect the KMS condition for the generating functional at the classical stochastic level, we require the corresponding dynamical KMS symmetry acting on the NG field as follows [40, 41]:

$$\begin{cases} \pi_R(t, \boldsymbol{x}) \to \pi'_R(t, \boldsymbol{x}) = \pi_R(-t, \boldsymbol{x}), \\ \pi_A(t, \boldsymbol{x}) \to \pi'_A(t, \boldsymbol{x}) = \pi_A(-t, \boldsymbol{x}) - i\beta \partial_t \pi_R(-t, \boldsymbol{x}), \end{cases} \tag{A.3}$$

where $\beta \equiv 1/T$ denotes the inverse temperature characterizing the initial thermal density. Note that this symmetry involves the temporal inversion, and as a result, it defines $\mathbb{Z}_2$ symmetry. Thus, in addition to three requirements introduced in Sec. 3.3, we assume that the effective action for the closed system domain wall remains invariant under the dynamical KMS transformation (A.3) as follows:

$$\text{KMS condition}: S_{\text{eff}}[\pi'_R, \pi'_A] = S_{\text{eff}}[\pi_R, \pi_A] + (\text{surface term}). \tag{A.4}$$

A remarkable property of the dynamical KMS symmetry is that it mixes the $A$-type field and the time-derivative of the $R$-type field. As a result, the effective action needs to contain them in a consistent manner. The dynamical KMS symmetry (A.4) guarantees the classical stochastic version of the KMS condition for the closed-time-path generating functional.

**Constructing the general effective Lagrangian.** Let us then write down the general effective Lagrangian in the classical stochastic limit. Here, we restrict ourselves to the leading-order result in the derivative expansion.

We start from the terms $O(\mathcal{A}^2)$. Owing to the shift symmetry for $\pi_A$, possible leading-order derivative terms are $(\partial_t \pi_A)^2$ and $(\partial_i \pi_A)^2$. We here neglect the former term because it has to be accompanied by the $O(p^3, \mathcal{A})$ term to satisfy the dynamical KMS symmetry, which is beyond $O(p^2)$ regime of our interest. One can also say that neglecting $(\partial_t \pi_A)^2$ term gives a consistent truncation with the dynamical KMS symmetry. On the other hand, the presence of $(\partial_i \pi_A)^2$ together with the KMS symmetry leads to an $O(\pi_A)$ term proportional to

$\partial_i \pi_A \partial_t \partial_i \pi_R = \partial_i \pi_A \partial_t \partial_i X_R$, whose coefficient is related with each other. In short, we find two terms

$$-\kappa(X_R)\partial_i \pi_A \partial_t \partial_i X_R + \mathrm{i}T\kappa(X_R)\partial_i \pi_A \partial_i \pi_A, \tag{A.5}$$

which represents the fluctuation-dissipation partner terms related by the KMS symmetry.

Let us now write down other $O(\mathcal{A})$ terms. Using the building blocks (A.2 ), we can construct all possible terms up to $O(p^2)$ terms. It is remarkable that the dynamical KMS symmetry also eliminates an apparently possible term $\gamma(X_R)\partial_t \pi_A$ since it does not respect the KMS symmetry. As a result, the leading-order effective Lagrangian in closed systems is identified as

$$\begin{aligned}
\mathcal{L}_{\mathrm{eff}} &= f_t(X_R)\partial_t X_R \partial_t \pi_A - f(X_R)\partial_i X_R \partial_i \pi_A - \frac{1}{2}f_s(X_R)[(\boldsymbol{\nabla} X_R)^2 - 1]\partial_i X_R \partial_i \pi_A \\
&\quad - \kappa(X_R)\partial_i \pi_A \partial_t \partial_i X_R + \mathrm{i}T\kappa(X_R)\partial_i \pi_A \partial_i \pi_A \\
&= f_t(x)\partial_t \pi_R \partial_t \pi_A - f_s(x)\partial_x \pi_R \partial_x \pi_A - \kappa(x)\partial_i \pi_A \partial_t \partial_i \pi_R + \mathrm{i}T\kappa(x)\partial_i \pi_A \partial_i \pi_A \\
&\quad - f(x)\partial_x \pi_A - f(x)\partial_i \pi_R \partial_i \pi_A - f'(x)\pi_R \partial_x \pi_A + O(\pi^3),
\end{aligned} \tag{A.6}$$

where we kept the quadratic fluctuation term in the second line. Note that the term proportional to $f(X_R)$ generates the tad-pole term, and thus, the elimination of that leads to $f(x) = \mathrm{const.}$[9] We emphasize that the closed system effective Lagrangian cannot support terms like $\gamma(X_R)$ and $\lambda(X_R)$ appearing in the open system counterpart. Thus, one sees that the dissipative term $\gamma(X_R)$ and the KPZ terms $\lambda(X_R)$ are peculiar to the symmetry broken state in the open system.

**Energy spectrum.** Based on the identified effective Lagrangian (A.6 ), we can immediately find the energy spectrum for the fluctuation. As in the main text, we demonstrate them in the two simple regimes; the thin-wall and thick-wall regimes.

Let us first start with the thin-wall regime. In the thin-wall regime, we have the dimensionally reduced effective Lagrangian given by

$$\begin{aligned}
\mathcal{L}_{\mathrm{thin}}^{(2)} &= \bar{f}_t \partial_t \tilde{\pi}_R \partial_t \tilde{\pi}_A - \bar{f}\partial_{i\perp}\tilde{\pi}_A \partial_{i\perp}\tilde{\pi}_R - \bar{\kappa}\partial_i \tilde{\pi}_A \partial_t \partial_i \tilde{\pi}_R + \mathrm{i}T\bar{\kappa}\partial_{i\perp}\tilde{\pi}_A \partial_{i\perp}\tilde{\pi}_A \\
&= \frac{\mathrm{i}}{2}\begin{pmatrix}\tilde{\pi}_R & \tilde{\pi}_A\end{pmatrix}\begin{pmatrix} 0 & \mathrm{i}[\bar{f}_t \partial_t^2 - \bar{f}\boldsymbol{\nabla}_\perp^2 + \bar{\kappa}\partial_t \boldsymbol{\nabla}_\perp^2] \\ \mathrm{i}[\bar{f}_t \partial_t^2 - \bar{f}\boldsymbol{\nabla}_\perp^2 - \bar{\kappa}\partial_t \boldsymbol{\nabla}_\perp^2] & -2T\bar{\kappa}\boldsymbol{\nabla}_\perp^2 \end{pmatrix}\begin{pmatrix}\tilde{\pi}_R \\ \tilde{\pi}_A\end{pmatrix},
\end{aligned} \tag{A.7}$$

where we introduced low-energy coefficients with overbar after performing $x$-integration of the corresponding coefficient functions. Investigating the pole of the retarded Green's function, we find the dispersion relation of the fluctuating domain wall as

$$\omega(\boldsymbol{k}_\perp) = \frac{\pm\sqrt{4\bar{f}\bar{f}_t \boldsymbol{k}_\perp^2 - \bar{\kappa}^2 \boldsymbol{k}_\perp^4} - \mathrm{i}\bar{\kappa}\boldsymbol{k}_\perp^2}{2\bar{f}_t} = \pm c_{s\perp}|\boldsymbol{k}_\perp| - \frac{\mathrm{i}}{2}D_\perp \boldsymbol{k}_\perp^2 + O(k^3), \tag{A.8}$$

where we introduced $c_{s\perp} \equiv \sqrt{\bar{f}/\bar{f}_t}$ and $D_\perp \equiv \bar{\kappa}/\bar{f}_t$ on the rightmost side. Note that we now have the propagating NG mode in closed systems in sharp contrast to the purely diffusive NG mode in open systems discussed in the main text.

We next consider the thick-wall regime. In this case, we can show the constant low-energy coefficient $f(x) = \mathrm{const.}$ vanishes with the help of the thermodynamic consideration. In fact, the term proportional to $f(x) = \mathrm{const.}$ leads to the linear term in the thermodynamic potential, which spoils the thermodynamic stability. Thus, if the system stays in a stable equilibrium state,

---

[9]Furthermore, the dynamical KMS symmetry also requires $f'(x) = 0$, which is equivalent to the condition from the elimination of the tad-pole term.

$f(x) = 0$ holds in the thick-wall regime. As a result, we obtain the leading-order effective Lagrangian as

$$
\begin{aligned}
\mathcal{L}_{\text{thick}}^{(2)} &= f_t \partial_t \pi_R \partial_t \pi_A - f_s \partial_x \pi_R \partial_x \pi_A - \kappa \partial_i \pi_A \partial_t \partial_i \pi_R + iT\kappa \partial_i \pi_A \partial_i \pi_A \\
&= \frac{i}{2} \begin{pmatrix} \pi_R & \pi_A \end{pmatrix} \begin{pmatrix} 0 & i[f_t \partial_t^2 - f_s \partial_x^2 + \kappa \partial_t \boldsymbol{\nabla}^2] \\ i[f_t \partial_t^2 - f_s \partial_x^2 - \kappa \partial_t \boldsymbol{\nabla}^2] & -2T\kappa \boldsymbol{\nabla}^2 \end{pmatrix} \begin{pmatrix} \pi_R \\ \pi_A \end{pmatrix}.
\end{aligned}
\tag{A.9}
$$

From the retarded Green's function, we obtain the dispersion relation anisotropic in the momentum space. Introducing the momentum in the cylindrical coordinate as $\boldsymbol{k} = (|\boldsymbol{k}|\cos\theta, \boldsymbol{k}_\perp)$, we identify the dispersion relation as

$$
\omega(\boldsymbol{k}) = \frac{\pm\sqrt{4 f_t f_s \boldsymbol{k}^2 \cos^2\theta - \kappa^2 \boldsymbol{k}^4}}{2 f_t} = \pm c_s |\boldsymbol{k}| \cos\theta - \frac{i}{2} D \boldsymbol{k}^2 + O(\boldsymbol{k}^3).
\tag{A.10}
$$

Note that the low-momentum behaviors of the spectrum are qualitatively different depending on its direction. In fact, one sees that the leading low-momentum behavior is linear ($\omega \sim k_x$) along the modulation direction while it is quadratic perpendicular to the modulation ($\omega \sim \boldsymbol{k}_\perp^2$).[10] This is a general feature of the effective field theory of one-dimensional modulating phase appearing in, e.g., the smectic-A phase of liquid crystals [98,99], the Fulde-Ferrell-Larkin-Ovchinnikov phase of superconductors [100,100,101], and the spiral phase of chiral magnets [102–107]. The obtained result gives a generalization of the anisotropic dispersion for the finite-temperature one-dimensional modulation phase.

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
