# Peer review of "Effective field theory of fluctuating wall in open systems: from a kink in Josephson junction to general domain wall"

_SciPost Physics, doi:SciPost Phys. 12, 160 (2022)_

## Round 2 · Referee Report · Anonymous (Referee 1) · 2021-12-30

Report

The paper introduces an effective field theory (EFT) of the Goldstone modes associated with the breaking of translations due to the presence of a domain wall with dissipative dynamics. Sec. 2 deals with the specific case of the Josephson junction, where the authors obtain the dispersion relations of the Goldstone modes in the thin-wall and thick-wall limits. After reviewing a general EFT approach for open systems, in sec. 4 the authors extend the analysis of sec. 2 to domain walls in general dimensions and propose an EFT for the Goldstone modes in the thin- and thick-wall limits that does not require knowing the profile of the domain wall, and that accounts for dissipation. Some properties of the dispersion relations predicted for the Goldstone modes are discussed and, in particular, it is speculated that the Kardar-Parisi-Zhang universality class might arise from some of these systems.

The paper has a clear structure, and it contains insightful results which are useful for further research on this topic. Therefore, I recommend the paper for publication, after the remarks below have been addressed.

Main points: - What is the reason for setting $\phi_A=0$ in finding the saddle point(s) of eq. (20)? - In sec. 4.3, the expansion in the "A-type" fields seems to coincide with the $\hbar$ expansion. This would be misleading as the two expansions have normally a different meaning (see e.g. sec. 4.1 of A. Kamenev - "Field Theory of Non-Equilibrium Systems"). I think the authors should simply refer to their counting as an expansion in A-type fields, not in $\hbar$. - Are the couplings in eq. (79) completely arbitrary? Given that (79) comes from expanding around a saddle-point, this does not seem straightforward. - Eq. (89): it is not a priori clear how the anisotropic terms $\lambda_s$ and $f_{tx}$ can survive in the thick-wall regime. In this regime, the Goldstone would seem to be insensitive to the structure of the domain wall, therefore one expects the effective dynamics to be isotropic. If the authors believe that (89) can be anisotropic, I think they should sketch a situation where this happens, since the analysis of the following two pages is contingent on this possibility.

Further comments/questions: - It would be good to mention the physical meaning of the mass term appearing in eq. (1). - Above eq. (3): I think the expression "noise-averaged" is a bit tricky here, since eq. (1) is nonlinear. It would probably be clearer to say "in the mean-field limit". - Eq. (25) contains the term $\pi_A \partial_t\nabla^2\pi_R$, while eq. (81), which is a generalization of (25), does not. Is this due to a truncation in the derivative expansion in (79)?

Finally, I found a few typos: - Eq. (27), $\nabla$ -> $\partial_y$ - p. 3, second paragraph: "dissipative effect" -> "dissipative effects" - p. 4, first paragraph: "induces the KPZ" -> "induce the KPZ" - p. 11, last line: "the momentum larger" -> "momentum larger than"

  • validity: -
  • significance: -
  • originality: -
  • clarity: -
  • formatting: -
  • grammar: -

Author:  Keisuke Fujii  on 2022-02-18  [id 2222]

(in reply to Report 1 on 2021-12-30)
Category:
answer to question

Reply to the Referee Report 1

Thank you very much for your careful reading as well as useful and insightful comments and questions. We have examined all the comments and questions. Our answers and modifications are indicated below. For your reference, we have also attached a manuscript in which the revised parts are colored.

Main points: 1) The generating functional satisfies Z=1 because of Eq. (11) in the absence of the external fields. This condition forces the MSR action to vanish when we evaluate it with a mean-field solution. On the other hand, substituting the first equation of Eq. (20) into the MSR action, we find it becomes int (A/2)phi_A^2. Thus, one finds the solution phi_A=0 to satisfy S_{MSR}=0. We added this explanation below Eq. (20).

2) We modified how to refer to our counting from the hbar expansion to an expansion with respect to A-type fields. In the manuscript, the explanation in sec. 4.3 was modified. Also, throughout the manuscript, the order counting was modified to be done in terms of mathcal{A} instead of hbar.

3) The coefficient functions in Eq. (79) are not completely arbitrary; their functional forms are, in principle, determined by the microscopic information of the system; in particular, the configuration of the inhomogeneous condensate bar{Phi}(x) is expected to determine the functional form of all coefficient functions. Also, the sign of the coefficient functions would be constrained from the stability of the steady state (Indeed, we assume the signs of the coefficients in the discussion of the thin/thick wall regime). However, we cannot determine the functional forms within the low-energy effective theory. We added these explanations below Eq. (79).

4) In the thick wall regime, we consider the slope of the wall to be a non-zero constant. This is the same as the thick wall regime in the model analysis, where the wall thickness m is not taken to be zero. In this regime, the isotropy is not completely recovered due to the non-zero slope, and the anisotropic terms lambda_s and f_{tx} can survive. It is true, however, that they do not appear easily, just as the anisotropy does not appear in the thick wall regime in the model analysis. These anisotropic terms break the discrete symmetry discussed in Sec 4.6, which exchanges the two different steady states separated by the domain wall. Therefore, they would not appear in the system enjoying the discrete symmetry. We added these explanations below Eq. (89). We also modified some sentences in section 4.6 to reflect the comments of the other referee.

Further comments/questions: 1) We added the explanation of the mass term below eq. (1).

2) We modified the sentence above eq. (3): "the noise-averaged Eq.~(1)" -> "Eq. (1) in the mean-field limit, where the right-hand side is replaced by its averaged value 0"

3) Yes, pi_{A}patial_tnabla^2pi_{R} is already truncated as a higher-order term in Eq.~(79). We mentioned it in footnote 5.

Other modifications: 1) Taking this opportunity, we also removed some typos throughout the paper, which does not change the contents of the paper. 2) Funding information was modified.

Sincerely yours, Keisuke Fujii Masaru Hongo

Attachment:

JJ_Domain-wall_and_Keldysh-EFT.pdf

---

## Round 2 · Referee Report · Anonymous (Referee 2) · 2022-1-28

Report

Dissipative systems can have symmetries for which the Noether currents do not have expectation values in any state and are therefore unobservable. This manuscript builds on the realization (due to Refs. [49-53]) that these symmetries can be spontaneously broken, with observable physical consequences. These symmetries are typically linearly realized on the MSR fields; see, e.g., Eq. (14). In contrast, nonlinearly realized symmetries such as shift symmetries (e.g. in Eq. (12), setting m=0) lead to regular, observable, Noether currents. In the present paper, the authors study such a broken symmetry situation relevant to a magnetic flux trapped in a 2d Josephson junction, where translations are spontaneously broken by a (thin or wide) flux line.

The paper is clearly written and accessible to a wide audience. Most of it seems technically correct, I therefore recommend its publication after a few issues are addressed:

  • Could the authors explain why it is not reasonable to expect fluctuation-dissipation relations to hold in these systems? Are they expected to be approximately satisfied or strongly violated?

  • One may be concerned that the parametrization (22) for \phi_R is not well defined, given that the image of \bar \phi is [0,2\pi].

  • In the thick wall regime, the authors find a diffusive mode that propagates in two dimensions. On the other hand, one can consider a system with approximate translation invariance that is spontaneously broken (this situation arises e.g. in unidirectional charge density waves in a variety of systems). In this latter situation, all modes are overdamped because momentum can leak out of the system (the dissipative properties of such systems has been the subject of recent interest, see e.g. [1702.05104, 1708.08306, 1908.01175]). What distinguishes these two situations? Can the authors accomadate in their framework additional terms that would lead to damping of the Goldstone? Why would these effects be absent in Josephson junctions?

  • There is a typo in Eq. (97) : the KPZ term should be cubic in fields.

  • The authors remark at the end of Sec. 4 that interesting nonlinear terms which may lead to KPZ dissipation are only allowed in the absence of reflection symmetry. However, given that rotation symmetry is assumed (even though it is spontaneously broken), I would expect that this implies that x_perp -> - x_perp symmetry would also be absent in this case. This would then allow bare \nabla_\perp terms e.g. in Eq. (85) -- I am focusing here on the situation where the domain wall has 1 spatial dimension, which is the situation where KPZ terms would be most interesting because relevant under RG. These additional terms in the Lagrangian would be more relevant than those considered, and would entirely change the scaling analysis. Can the authors comment on this?

  • validity: high
  • significance: high
  • originality: high
  • clarity: top
  • formatting: -
  • grammar: -

Author:  Keisuke Fujii  on 2022-02-18  [id 2223]

(in reply to Report 2 on 2022-01-28)
Category:
answer to question

Reply to the Referee Report 2

Thank you very much for your careful reading as well as useful and insightful comments and questions. We have examined all the comments and questions. Our answers to them and modifications are indicated below. For your reference, we have also attached a manuscript in which the revised parts are colored.

1) If the environment, composed of electrons in the superconducting electrodes for the Josephson junction, is in thermal equilibrium, the fluctuation-dissipation relation is expected to hold. Indeed, we expect it to hold in the experimental setup of the Josephson junction. The reason why we do not assume the fluctuation-dissipation relation is mainly due to the theoretical purpose to perform a general analysis, in the same manner with the second part (Secs. 3-4) of our paper. We modified the sentence below eq. (2) to make this clearer.

2) The variable phi (phi_R) denotes a phase difference between two superconducting electrodes, and thus, takes the value in [0,2pi]. In the revised manuscript, we explicitly mentioned a range of phi above eq. (1) and that of phi_R below eq. (6).

3) Thank you for pointing out interesting previous studies. The symmetry specific to open systems, which we refer to as the A-type one in the manuscript, is different from the approximate symmetry that is explicitly but weakly broken. As discussed in section 3.2, the dissipation and noise explicitly break the R-type symmetry, but not the A-type symmetry, and we focus on the Goldstone modes associated with the spontaneous breaking of the remaining A-type symmetries. On the other hand, the references that the referee suggested discuss pseudo-Goldstone modes associated with the spontaneous broken approximate translational symmetry.

In this case, not only R-type symmetry but also A-type one is explicitly broken. This is the difference between our setup and those discussed in their papers. In our case, Goldstone modes are associated with the spontaneous breaking of the exact A-type symmetry, and thus, they show a gapless diffusive behavior. On the other hand, pseudo-Goldstone modes are associated with the spontaneous breaking of the approximate A-type symmetry, so that they show a gapped overdamped behavior.

Can the authors accommodate in their framework additional terms that would lead to damping of the Goldstone? Why would these effects be absent in Josephson junctions?

Yes, our formulation can cover their situation by adding an additional A-type symmetry breaking terms to the effective Lagrangian. The Josephson junction system that we discussed enjoys the exact A-type symmetry, so that it does not lead to a gapped relaxational mode. Based on the referee's comment, we modified the manuscript to mention approximate symmetry at the end of the third | paragraph on page 3.

4) We corrected the typo.

5) What we called the inversion symmetry in the previous manuscript refers to the symmetry that exchanges the two different steady states separated by the domain wall. Thus, this symmetry is not a spatial inversion symmetry, but rather the discrete symmetry acting on fields (In the case of the Josephson junction system, this symmetry is realized as the transformation phi_Rto 2pi-phi_R and phi_Ato -phi_A. In a general setup, this transformation acts on our dynamical variables as X_R to -X_R and pi_Ato -pi_A, as described in the manuscript). Therefore, even in the absence of the discrete exchanging symmetry, we still assume rotational symmetry within the plane along the wall, and thus a term with a single nabla_{perp} cannot appear.

We find that the terminology "inversion symmetry" is confusing with spatial inversion symmetry, so that we changed it to call the discrete symmetry, and added an explanation in the last paragraph of Sec. 4.6.

Other modifications: 1) Taking this opportunity, we also removed some typos throughout the paper, which does not change the contents of the paper. 2) Funding information was modified.

Sincerely yours, Keisuke Fujii Masaru Hongo

Attachment:

JJ_Domain-wall_and_Keldysh-EFT_AONFvJn.pdf

---

## Round 3 · Referee Report · Anonymous · 2022-4-1

Report
The points in my previous report have been addressed. I am happy to recommend the paper for publication.

---

## Round 3 · Referee Report · Anonymous · 2022-4-17

(Invited Report)Report
The authors have addressed my questions and comments. I recommend this manuscript for publication.

---

## Round 3 · List of Changes

We modified some sentences to reflect the comments of the referees.

You are currently on this page

---

## Editorial Decision

published